# Stromal heterogeneity may explain increased incidence of metaplastic breast cancer in women of African descent

Brijesh Kumar[1,8,9], Aditi S. Khatpe[1,2,9], Jiang Guanglong [3], Katie Batic[1], Poornima Bhat-Nakshatri[1], Maggie M. Granatir[4], Rebekah Joann Addison[4], Megan Szymanski[4], Lee Ann Baldridge[4], Constance J. Temm[4], George Sandusky [4], Sandra K. Althouse[5], Michele L. Cote[6], Kathy D. Miller[3], Anna Maria Storniolo [3] & Harikrishna Nakshatri [1,2,7] ✉

The biologic basis of genetic ancestry-dependent variability in disease incidence and outcome is just beginning to be explored. We recently reported enrichment of a population of ZEB1-expressing cells located adjacent to ductal epithelial cells in normal breasts of women of African ancestry compared to those of European ancestry. In this study, we demonstrate that these cells have properties of fibroadipogenic/mesenchymal stromal cells that express PROCR and PDGFRα and transdifferentiate into adipogenic and osteogenic lineages. PROCR + /ZEB1 + /PDGFRα+ (PZP) cells are enriched in normal breast tissues of women of African compared to European ancestry. PZP: epithelial cell communication results in luminal epithelial cells acquiring basal cell characteristics and IL-6-dependent increase in STAT3 phosphorylation. Furthermore, level of phospho-STAT3 is higher in normal and cancerous breast tissues of women of African ancestry. PZP cells transformed with HRas$^{G12V}$ ± SV40-T/t antigens generate metaplastic carcinoma suggesting that these cells are one of the cells-of-origin of metaplastic breast cancers.

Population-based research consistently reports that African American women have the highest incidence and mortality rates from triple negative breast cancer (TNBC) compared other racial and ethnic groups despite adjustment for socioeconomic and other contributing factors[1]. Even in cases of estrogen receptor positive (ER+)/HER2-/node negative breast cancers with intermediate Oncotype recurrence scores[2], clinical outcome is worse in African American women compared to non-Hispanic White women[3]. Furthermore, ductal carcinoma in situ (DCIS), which is typically considered to be of minimal risk of recurrence, tends to be more deleterious to African American women compared to non-Hispanic White women[4]. Additionally, disparities in pathologic complete response among patients receiving neoadjuvant therapy has been reported, with lower response rates in Black women[5]. These studies have all relied on self-reported race and ethnicity, which are not biologically-defined, but are instead social constructs that have changed over time[6]. Martini et al. went beyond these classifications and utilized biological markers associated with country of origin and reported distinct immunologic landscapes within TNBC tumors from

[1]Department of Surgery, Indiana University School of Medicine, Indianapolis, IN 46202, USA. [2]Department of Biochemistry and Molecular Biology, Indiana University School of Medicine, Indianapolis, IN 46202, USA. [3]Department of Medicine, Indiana University School of Medicine, Indianapolis, IN 46202, USA. [4]Department of Pathology and Laboratory Medicine, Indiana University School of Medicine, Indianapolis, IN 46202, USA. [5]Department of Biostatistics and Health Data Science, Indiana University School of Medicine, Indianapolis, IN 46202, USA. [6]Richard M. Fairbanks School of Public Health, Indiana University, Indianapolis, IN 46202, USA. [7]VA Roudebush Medical Center, Indianapolis, IN 46202, USA. [8]Present address: School of Biomedical Engineering, Indian Institute of Technology (Banaras Hindu University), Varanasi, UP 221005, India. [9]These authors contributed equally: Brijesh Kumar, Aditi S. Khatpe. ✉e-mail: hnakshat@iupui.edu

women of African ancestry (AA) compared to women of European ancestry (EA), with further differences within regional African ancestry[7]. We had previously reported distinct differences in immune cells within tumor adjacent normal tissues of women of African compared to European ancestry[8]. Ancestry driven differences in tumor mutation burden, which impact clinical outcome from immune checkpoint inhibitor treatment, have also been reported[9].

Recently more studies have focused on understanding the intersection between biology and socioeconomic aspects of breast cancer disparities that contribute to molecular differences in tumor characteristics, as there is increasing evidence for disparities persisting even when access to health care is equal[10–12]. Higher percentage of African ancestry and lower neighborhood socioeconomic status are associated with higher mortality further suggesting an interplay between ancestry-dependent biology and socioeconomic status in human health[13]. A breast cancer specific study demonstrated that while 19% of the disparity can be attributed to neighborhood disadvantage and insurance status, tumor biological characteristics contribute to 20% of disparity[14]. Another study of hormone receptor positive breast cancers showed that even after adjusting for neighborhood-deprivation index and insurance, identifying as Black race is associated with shorter relapse-free survival[15]. Circulating Interleukin-6 (IL-6) has emerged as a major measure of socioeconomic stress in women but not men[16] and could potentially integrate genetic ancestry-dependent differences in biology with socioeconomic stress. With the recent discovery of the presence of an extra 296,485,284 base pairs DNA in populations of African descent affecting 315 distinct protein-coding genes[17], some of the differences in tumor biology based on ancestry may come from differences in the normal tissue biology.

To discover genetic ancestry-dependent differences in normal breast biology, we initiated studies using healthy breast tissues from women of different genetic ancestry (termed Normal-Healthy hereafter) donated to the Susan G. Komen Tissue Bank (KTB) at the Indiana University Simon Comprehensive Cancer Center (IUSCCC). We had previously demonstrated that cultured cells from breast tissues of women of AA are enriched for Protein C Receptor-positive/Epithelial Cell Adhesion Molecule-negative (PROCR+/EpCAM−) and CD44+/C24− cells compared to women of EA[18]. In the mouse mammary gland, the PROCR+/EpCAM− cells have been demonstrated to function as multi-potent stem cells[19]. Subsequently, we demonstrated that the normal breasts of women of AA contain elevated numbers of Zinc Finger E-box binding homeobox 1-positive (ZEB1+) cells that surround the ductal epithelial cells compared to women of EA[8]. Stromal ZEB1 expression in breast cancer inversely correlates with abundance of multiple immune cell types[20]. ZEB1 expression has been linked to claudin-low and metaplastic breast cancer subtypes, both of which are more common in women of AA compared to women of EA[21–24]. These observations prompted us to further characterize stromally located ZEB1+ cells for their potential role in breast cancer initiation and progression within the context of genetic ancestry.

In this study, we report the establishment of several ZEB1+ cell lines, all derived from the breast tissues of women of AA. These cells express PROCR and Platelet-derived Growth Factor Receptor alpha (PDGFRα), similar to the stromal adipogenic progenitors cells of the mouse mammary gland that trans-differentiate into multiple cell types[25]. Based on cell surface expression of PROCR and PDGFRα combined with ZEB1 expression, we have named these cells as 'PZP cells' for simplicity. Under appropriate growth conditions, these cells differentiate into adipogenic or osteogenic lineages and express the adipogenic marker Peroxisome Proliferator Activated Receptor gamma (PPARγ). Cell surface marker profiles reveal similarity of these cells to human prostate-derived mesenchymal stem cells (CD44+/CD90+/CD73+/CD105+)[26]. PZP-breast epithelial cell communication results in elevated expression of IL-6 in PZP cells. Furthermore, epithelial cells acquire CD49f+/EpCAM− basal cell characteristics, which

have been shown to increase with aging and to increase susceptibility to breast cancer[27,28]. PZP cells, when transformed with oncogenes, generate metaplastic carcinomas in NOD/SCID Gamma (NSG) mice suggesting that these cells are one of the cells-of-origin of metaplastic carcinomas of the breast[29].

## Results

### Generation of <u>P</u>ROCR+/<u>Z</u>EB1+/<u>P</u>DGFRα+ (PZP) cell lines from breast tissues of women of African ancestry

We created nine immortalized PZP cell lines from breast tissues of women of AA (KTB40, KTB42, KTB32, KTB53, KTB57, KTB59, KTB104, KTB106, and KTB109181) by overexpressing human telomerase gene (hTERT) using primary cells isolated and propagated from core breast biopsies. Enrichment of AA informative markers in these donors was confirmed through genotyping (Fig. 1a, and Fig. S1a). To phenotypically characterize and document heterogeneity, these cells were subjected to flow cytometry using various epithelial, stem, mesenchymal, and fibroblast markers. These cell lines were predominantly PROCR+/EpCAM− (Fig. 1b, c, and Fig. S1b−d). To further characterize stem cell-related gene expression, we compared these immortalized variants with the immortalized PROCR±/EpCAM+ luminal/basal cell variants from women of AA and EA we described previously[30]. PROCR+/EpCAM− cell lines expressed significantly higher levels of ZEB1 compared to PROCR±/EpCAM+ (KTB34 and KTB39) breast epithelial cell lines (Fig. 1d, and Fig. S1e). The gene expression pattern in KTB34 cell line from woman of EA overlaps with the Luminal-A intrinsic subtype of breast cancer, whereas gene expression in KTB39 from woman of AA overlaps with Normal-like intrinsic subtype of breast cancer[30].

A recent study described PDGFRα+ stromal cells as adipogenic progenitors of the mammary gland that trans-differentiate into epithelial cells and migrate into the duct when stimulated by Platelet-derived growth factor-C (PDGF-C)[25]. Interestingly, these cells also express PROCR[25]. We examined whether human breast PROCR+/ZEB1+ cells express PDGFRα. Indeed, >70% of cells were PDGFRα+ (Fig. 1e, f; Fig. S1f−h), and these cells also expressed PDGFRβ (Fig. S1i). CD49fhigh/EpCAMlow, CD49fhigh/EpCAMmedium, and CD49flow/EpCAMhigh cells are described as breast basal, luminal progenitor, and mature/differentiated epithelial cells, respectively[31]. None of the PZP cell lines were positive for CD49f and EpCAM (Fig. S2a). Morphologically, PZP cell lines showed features of mesenchymal stem and fibroblast like cells, which are loosely attached to cell culture plates, unlike epithelial cells (Fig. S2b).

### Trans-differentiating properties of PZP cells

Both adipogenic progenitors described in the mouse mammary gland[25] and fibroadipogenic progenitors (FAPs) described in the skeletal muscle[32] are able to undergo trans-differentiation into adipogenic and osteogenic lineages. In order to determine to what extent PZP cells show similarity to these other progenitors, PZP cells were subjected to adipogenic and osteogenic differentiation growth conditions. Indeed, PZP cells differentiated into adipocytes (Fig. 2a) and the lipid content was significantly increased in the differentiated PZP adipocytes (Fig. 2b). The PZP differentiated cells expressed adipocyte differentiation marker PPARγ (Fig. 2c). Adipogenic trans-differentiation capability of PZP cells was further confirmed through RUNX1 expression analysis (Fig. 2d). The PZP cells cultured in osteogenic media for 21 days showed osteogenic differentiation as evident from mineralization of matrix with $Ca^{2+}$ and positive alizarin red staining (Fig. 2e, f). Thus, PZP cells enriched in the breasts of women of AA could correspond to multipotent cells that can trans-differentiate into different cell types based on environmental cues.

### PZP cells express lobular fibroblast markers

In breasts, loose connective tissue is unique to the terminal duct lobular units (TDLUs), which drain into dense connective tissue

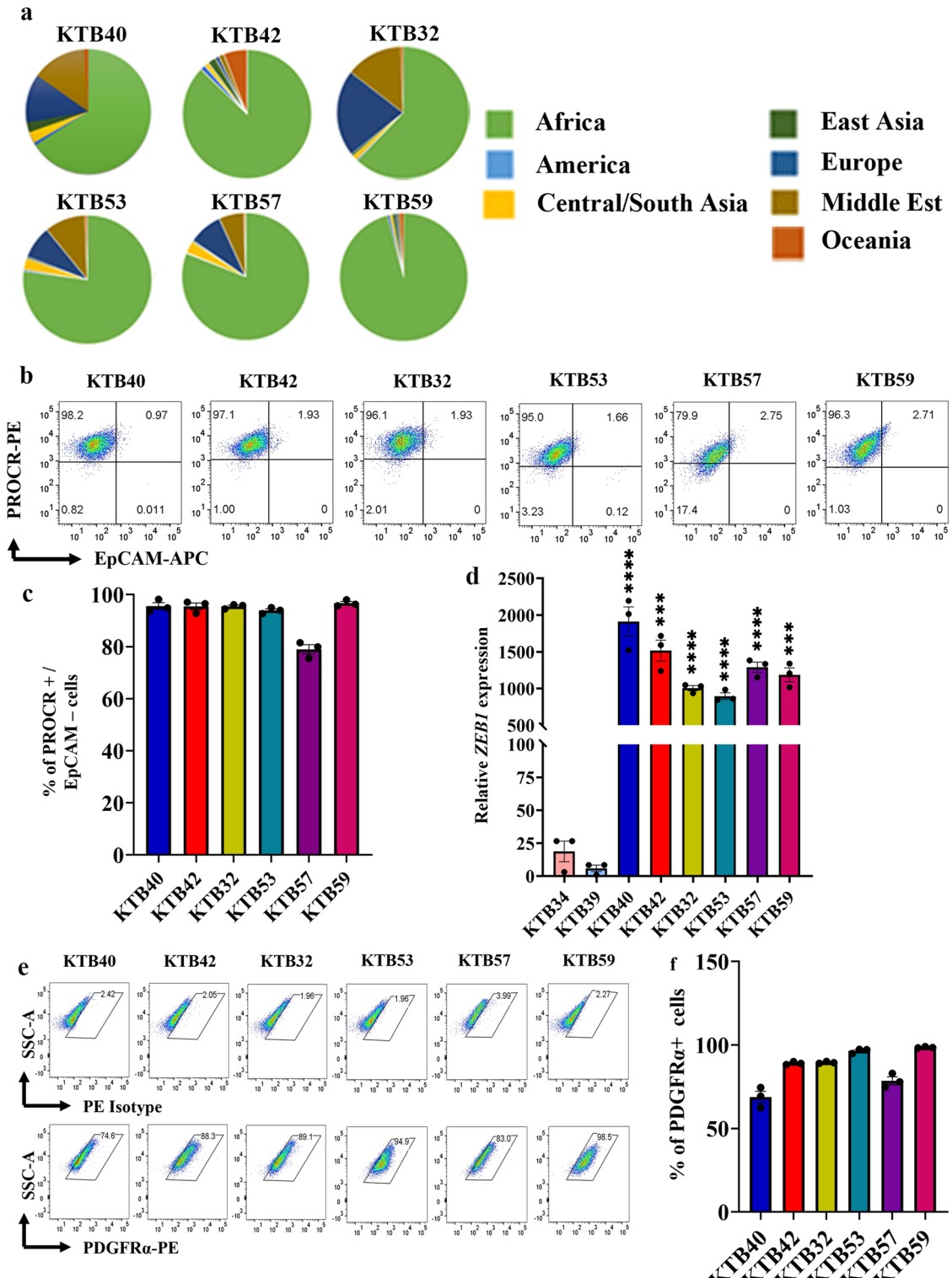

Fig. 1 | Establishment of PROCR⁺/ZEB1⁺/PDGFRα⁺ (PZP) cell lines from the
Normal-Healthy breast tissues of women of African ancestry. a Genetic ancestry
mapping of breast tissue donors (KTB40, KTB42, KTB32, KTB53, KTB57, and
KTB59) using a panel of 41-SNP. b PROCR⁺/EpCAM⁻ cells are enriched in established
cell lines from women of African ancestry (n = 3). Isotype controls are shown in
Fig. S1d. c Quantitation of PROCR⁺/EpCAM⁻ cells (n = 3). d ZEB1 expression levels in
various PROCR⁺/EpCAM⁻ cell lines compared to EpCAM⁺ (KTB34 and KTB39)
luminal cell lines (n = 3). KTB40*p < 0.0001, KTB42*p = 0.0002, KTB32*p < 0.0001,
KTB53*p < 0.0001, KTB57*p < 0.0001, KTB59*p = 0.0003. e PROCR⁺/ZEB1⁺ cell lines
express PDGFRα as determined by flow cytometry (n = 3). f Quantitation of
PDGFRα⁺ cells (n = 3). ***p < 0.001, ****p < 0.0001 by one-way ANOVA. All the data
points are shown as mean ± SEM. Source data are provided as a Source Data file.

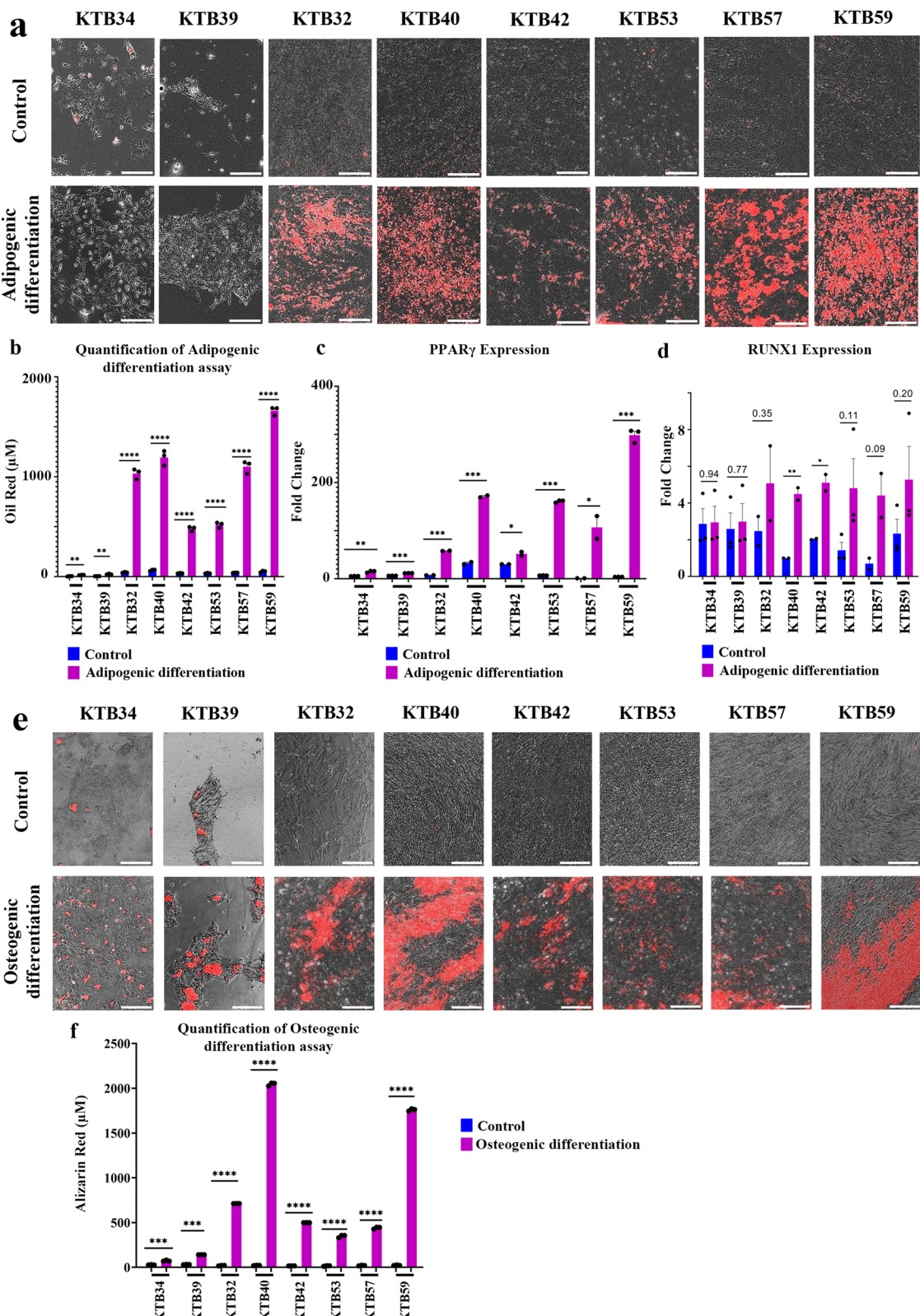

embedded interlobular ducts[33]. CD105[high] TDLU-resident lobular fibroblasts display properties different from interlobular fibroblasts[34]. While the CD105[high] lobular fibroblasts resemble mesenchymal stem cells (MSCs) both by phenotype and function, CD26[high] interlobular cells remain fibroblast restricted[34]. CD105[high]/CD26[low] and CD105[low]/CD26[high] lineages are considered to represent lobular and interlobular human breast fibroblastic cells (HBFCs), respectively[35]. To further characterize PZP cell lines, we examined the CD105 and CD26 staining

pattern. PZP cells are enriched for CD105[high]/CD26[low] population with inter-individual variability in the ratio between CD105[high]/CD26[-] and CD105[high]/CD26[low] cells (Fig. 3a–c, and Fig. S2c, d), which suggest that PZP cells reside predominantly in the lobules of breasts.

CD90[-]/CD73[high] and CD90[high]/CD73[high] cells are described as rare endogenous pluripotent somatic stem cells and potential mesenchymal stem cells, respectively[36]. PZP cells contained CD90[-]/CD73[high] and CD90[high]/CD73[high] subpopulations, with notable inter-individual

**Fig. 2 | Trans-differentiating properties of PZP cells. a** PZP cells undergo adipogenic differentiation under appropriate growth condition. Neural lipids stain red upon Oil Red-O staining (n = 3). Epithelial cell lines did not undergo adipogenic differentiation. **b** Quantification of lipid levels in control and differentiated PZP adipocytes (n = 3). (KTB34*p = 0.0021, KTB39*p = 0.0023, KTB32*p < 0.0001, KTB40*p < 0.0001, KTB42*p < 0.0001, KTB53*p < 0.0001, KTB57*p < 0.0001, KTB59*p < 0.0001.) **c** Adipogenic differentiated PZP cells show elevated level of adipocyte differentiation marker PPARγ (n = 3). (KTB34*p = 0.0012, KTB39*p < 0.0001, KTB32*p < 0.0001, KTB40*p = 0.0004, KTB42*p = 0.0311, KTB53*p < 0.0001, KTB57*p = 0.0472, KTB59*p < 0.0001.) **d** Adipogenic differentiation of PZP cells was confirmed through RUNX1 overexpression (n = 3). (KTB40*p = 0.0093, KTB42*p = 0.0209.) **e** PZP cells undergo osteogenic differentiation under appropriate growth condition. Mineralization of matrix with Ca²⁺ was visualized by alizarin red staining (n = 3). **f** Quantification of alizarin red staining in control and differentiated PZP osteocytes (n = 3). (KTB34*p = 0.0003, KTB39*p < 0.0001, KTB32*p < 0.0001, KTB40*p < 0.0001, KTB42*p < 0.0001, KTB53*p < 0.0001, KTB57*p < 0.0001, KTB59*p < 0.0001.) *p < 0.05, **p < 0.01, ***p < 0.001, ****p < 0.0001. The data was analyzed using a two-tailed t-test. All the data points are shown as mean ± SEM. Source data are provided as a Source Data file.

variability in the ratio between CD90⁻/CD73^high, CD90^low/CD73^high, CD90^med/CD73^high and CD90^high/CD73^high populations (Fig. 3d–g; Fig. S2e–h). CD44 and CD24 are the "original" markers used to characterize cancer stem cells (CSCs) in breast cancer[37]. Interestingly, PZP cells displayed CD44^high/CD24^low phenotype (Fig. 3h, i; Fig. S2i, j). However, lack of EpCAM expression in PZP cells suggest that they are not the "normal counterparts" of cancer stem cells. CD10 marker is used to isolate myoepithelial cells, although a recent study showed CD10 positivity of cancer associated fibroblasts[38,39]. PZP cells showed CD10⁺ phenotype (Fig. 3j, k; Fig. S2k and l).

CD44⁺/CD24⁻ and Aldehyde Dehydrogenase+ (ALDH⁺) cancer cells of the breast have been described as mesenchymal and epithelial-like breast cancer cells, respectively, and gene expression in these cells overlap with basal and luminal epithelial cells of the normal breast[40]. While PROCR⁺/EpCAM⁺ epithelial cell lines (KTB34 and KTB39) contained ALDH⁺ cells, as measured through ALDEFLUOR assay, none of the PZP cell lines contained significant levels of ALDH⁺ cells (Fig. S3a, b).

### PZP cells are enriched in the normal breasts of women of African compared to European ancestry

In our previous study, we had demonstrated that ZEB1+ cells in the normal breast are located in the stroma with close proximity to ductal epithelial cells[8]. We generated a tissue microarray (TMA) comprising healthy breast tissues from 49 and 154 women of AA and EA, respectively. The TMA was first created based on self-reported Race/ancestry and subsequently linked to genetic ancestry. Ancestry marker distribution pattern of donors is shown in Fig. S4a, b and details of ancestry analysis are provided in Materials and Methods. We used recently suggested race and ethnicity reporting guidelines to describe study populations[41]. AA marker levels ranged from 30 to 99% in self-reported Black women. EA marker levels ranged from 20-98% in self-reported non-Hispanic White women. The TMA was analyzed for protein levels of PROCR, ZEB1 and PDGFRα by immunohistochemistry (IHC). Because of our previous observation of elevated ZEB1⁺ cells in the non-tumor adjacent to tumor tissues (NATs) compared to breast tissues from healthy donors of European ancestry[8], the TMA also contained paired breast tumors and NATs of self-reported race (African American and White). However, for simplicity and to avoid confusion, NATs and tumors from self-reported African American and White women are labelled as AA and EA in figures and figure legends. To distinguish tissues of clinically healthy donors from NATs, clinically healthy tissues are labelled as Normal-Healthy. Clinicopathologic features of tumors are described in Table S1. Representative IHC staining patterns of PROCR in Normal-Healthy, NATs, and tumors are shown in Fig. 4a, b and statistical analyses are presented in Fig. 4c–e and in Table S2. PROCR expression was observed in both ductal and stromal cells (Fig. 4a, b) with ~5-fold higher expression in stromal cells compared to epithelial cells (Fig. S5a). PROCR-expressing cells were enriched in the Normal-Healthy breast tissues of women of AA compared to EA (Fig. 4a–c; Table S2). PROCR H-Score remained significantly higher in women of AA compared to EA when the analysis included only those with >75% African and >75% European ancestry markers (H-score: 56.41 in 17 AA vs 21.97 in 70 EA, p value < 0.0001). Furthermore,

PROCR H-Score positively correlated with African ancestry proportions (Coefficient 0.347, p = 2.75E−06) and negatively with European ancestry proportions (Coefficient −0.265, p = 4.00E−04). NATs of women of AA and EA contained significantly higher levels of PROCR⁺ cells compared to Normal-Healthy suggesting the field effect of tumors on PROCR expression and NATs are not "normal" (Fig. 4d). Tumors of women of EA displayed a modest increase in PROCR⁺ cells compared with those of NATs, while in women of AA, there were no differences between NATs and tumors (Fig. 4e). Thus, PROCR⁺ cells are intrinsically higher in the normal breasts of women of AA.

Consistent with our previous report regarding ZEB1 expression[8], Normal-Healthy breast tissues of women of AA displayed higher levels of ZEB1 H-score compared to women of EA. (Fig. 5a–c; Table S2). ZEB1 H-Score negatively correlated with EA proportions (Coefficient −0.208, p = 6.45E−03). ZEB1 H-Score was not significantly different between AA and EA women when the analysis included only those with >75% African and >75% EA ancestry markers, possibly due to small sample size (H-score: 1.66 in 13 AA vs 1.37 in 83 EA, p value 0.9658). NATs of women of both AA and EA contained significantly higher levels of ZEB1⁺ cells compared to corresponding Normal-Healthy (Fig. 5d). Tumors of women of EA displayed higher levels of ZEB1⁺ cells compared to NATs (Fig. 5e).

We examined the ZEB1 expression with respect to income level as a surrogate marker of socioeconomic stress (<20 K, 20–50 K, 50–100 K and >100 K in $), but did not observe any significant differences based on income levels (Fig. S5b, c). While this is an incomplete measure of socioeconomic stress, it supports the findings that the normal breasts of women of AA exhibit intrinsically higher levels of ZEB1⁺ cells but ZEB1⁺ cell numbers increase in the breast with cancer in women of EA, consistent with our previous report[8].

PDGFRα expression in the Normal-Healthy breast tissues was low and mostly in stromal cells (Fig. 6a and Fig. S5d). However, tumor epithelium showed some degree of positivity. PDGFRα-expressing cells were enriched specifically in the Normal-Healthy breast tissues of women of AA compared to women of EA (Fig. 6a–c; Table S2). PDGFRα H-Score remained significantly higher in women of AA compared to women of EA when the analysis included only those with >75% African and >75% European ancestry markers (H-score: 38.62 in 16 AA vs 13.41 in 87 EA, p value 0.0027). The PDGFRα H-Score positively correlated with AA proportions (Coefficient 0.484, p = 1.62E−10) and negatively with EA proportions (Coefficient −0.404, p = 1.74E−07). Normal-Healthy breast tissues contained significantly higher levels of PDGFRα⁺ cells compared to NATs only in women of AA, while no difference between Normal-Healthy breasts and NATs of women of EA was observed (Fig. 6d). Furthermore, we did not observe a significant difference in expression between NATs and tumors of women of AA (Fig. 6e), suggesting that PDGFRα⁺ cells are intrinsically higher in the normal breasts of women of AA. Taken together, our data suggest that PZP cells are elevated in the normal breasts of women of AA compared to women of EA, although expression of PROCR and PDGFRα in epithelial cells makes data interpretation a bit difficult.

We next examined the same TMA for luminal epithelial markers to rule out possible bias in our observations. Because FOXA1 along with another pioneer factor GATA3 and ERα form a lineage-restricted

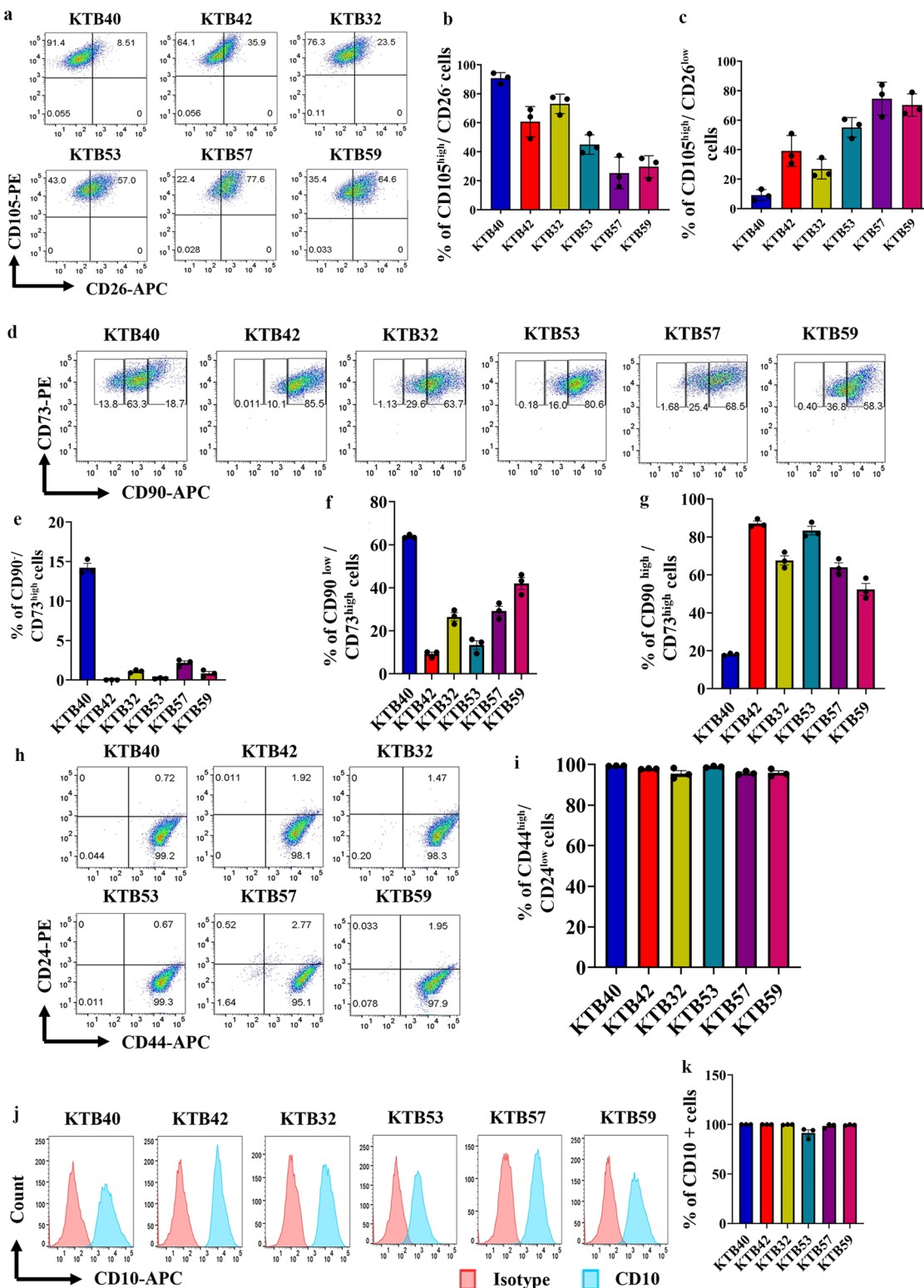

**Fig. 3 | Phenotypic characterization of PZP cells. a** PZP cell lines were stained with CD105 and CD26 antibodies to identify the lobular and interlobular origin of PZP cells ($n = 3$). Isotype controls are shown in Fig. S2a and e. **b** and **c** Quantification of CD105$^{high}$/CD26 and CD105$^{high}$/CD26$^{low}$ population of cells ($n = 3$). **d** PZP cell lines were stained with CD90 and CD73 antibodies to identify rare endogenous pluripotent somatic stem cells and potential mesenchymal stem cells ($n = 3$). Isotype controls are shown in Fig. S2. **e–g** Quantification of CD90$^-$/CD73$^{high}$, CD90$^{low}$/ CD73$^{high}$, and CD90$^{high}$/CD73$^{high}$ population of cells ($n = 3$). **h** PZP cell lines were stained with CD44 and CD24 antibodies to determine whether their phenotype overlaps with cancer stem cells ($n = 3$). Isotype controls are shown in Fig. S2. **i** Quantification of CD44$^{high}$/CD24$^{low}$ population ($n = 3$). **j** PZP cell lines were stained with CD10 antibody to determine overlap with myoepithelial cell marker expression ($n = 3$). **k** Quantification of CD10$^+$ population ($n = 3$). All the data points are shown as mean ± SEM. Source data are provided as a Source Data file.

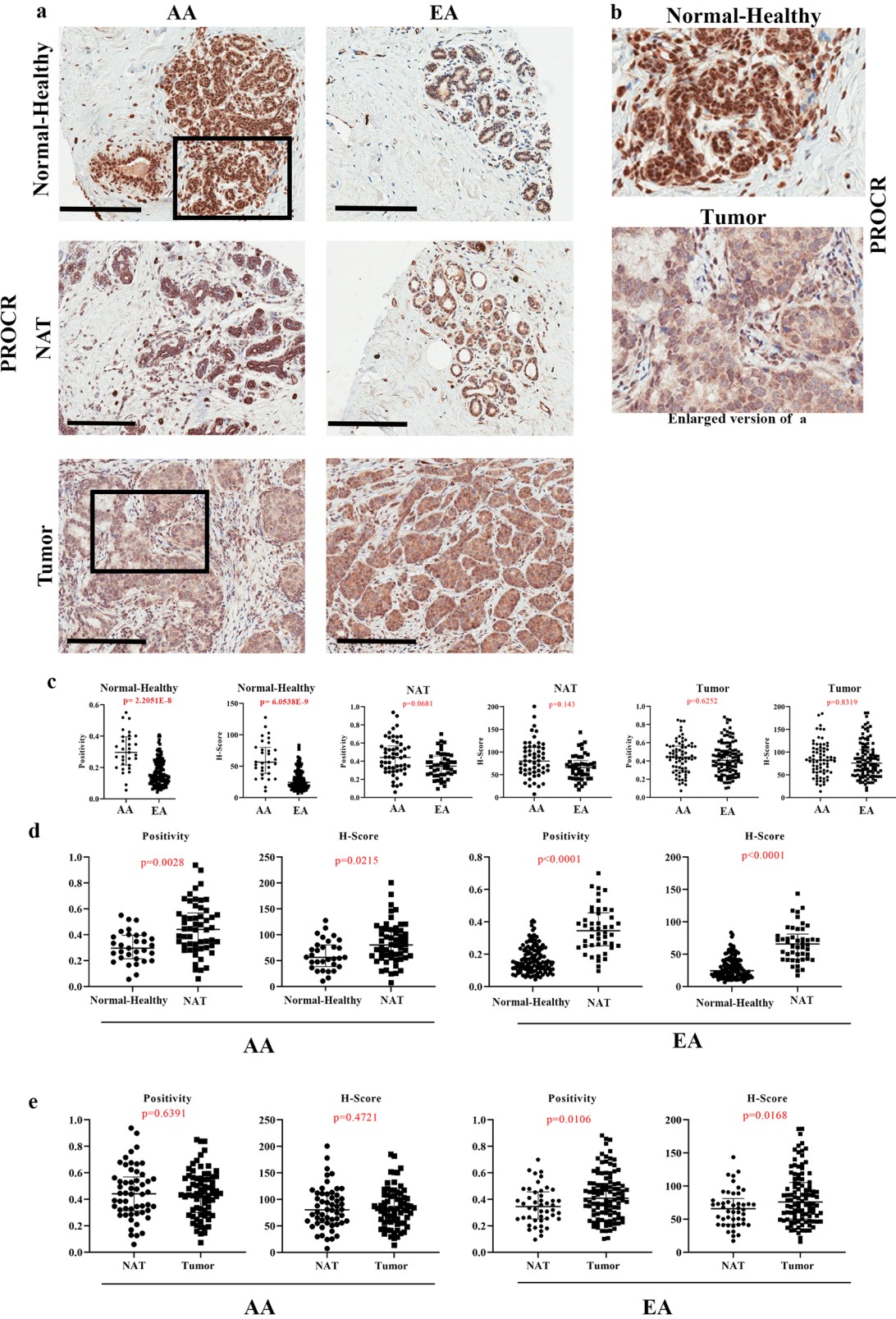

hormone-responsive signaling network in the normal breast[42], we examined the expression levels of these markers. Representative staining patterns of ERα, GATA3, and FOXA1 are shown in Fig. S6, Fig. S7, and Fig. S8, respectively. We did not observe genetic ancestry-dependent differences in ERα levels in Normal-Healthy tissues (Fig. S6a−c). NATs of women of AA contained modestly higher levels of ERα⁺ cells compared with those of Normal-Healthy, while this

difference was not observed in NATs of women of EA (Fig. S6d). Tumors of women of EA contained significantly higher levels of ERα⁺ cells compared with those of NATs, while this difference was not observed in women of AA (Fig. S6e). This is consistent with pathologic features of tumors in women of EA as the majority of them were ER⁺ tumors (Table S1, 75% ER⁺ in EA vs 60% ER⁺ in AA; 8% TNBCs in EA vs 16% TNBCs in AA). Similar to ERα, GATA3 expression in the normal breasts

**Fig. 4 | PROCR expression pattern in Normal-Healthy, NATs, and breast tumors.**
**a** Representative IHC of PROCR in Normal-Healthy, NATs, and/or tumors of women of AA and EA. **b** Enlarged view of PROCR expression in Normal-Healthy and tumor. **c** Differences in PROCR expression (positivity and H-score) between Normal-Healthy tissues of women of AA and EA. Differences in PROCR expression (positivity *p = 2.2051E−8 and H-score *p = 6.0538E−9) between Normal-Healthy tissues of AA and EA. The statistical test that was utilized to analyze the data was two-sided Wilcoxon Test. **d** Differences in PROCR expression (positivity and H-score) between Normal-Healthy and NATs of women of AA and EA (positivity *p = 4.588E−11 and

H-score *p = 1.4572E−11). The data were analyzed using two-sided Wilcoxon test. **e** Differences in PROCR expression (positivity and H-score) between NATs and tumors of women of AA and EA. Data were analyzed using two-sided Wilcoxon test. (Normal-Healthy-AA (n = 31), and EA (n = 129); NAT -AA (n = 35) and EA (n = 31); Tumor -AA (n = 41) and EA (n = 62)). Depending on availability, two cores of same NAT and tumor were included in the TMA. Dot blots in this and subsequent relevant figures include values from both cores of the same sample. All the data points are shown as mean ± SD. Source data are provided as a Source Data file.

did not show genetic ancestry-dependent differences (Fig. S7a–c). NATs of women of AA contained significantly higher levels of GATA3[+] cells compared with those of Normal-Healthy, while this difference was not observed in women of EA (Fig. S7d). Tumors of women of EA contained significantly higher levels of GATA3[+] cells compared with those of NATs, which is consistent with the clinical features of tumors, while this difference was not observed in women of AA (Fig. S7e). FOXA1 also did not show any genetic ancestry-dependent differences in Normal-Healthy tissues (Fig. S8a–c). NATs of women of AA and EA contained significantly higher levels of FOXA1[+] cells compared with those of Normal-Healthy (Fig. S8d). As expected, tumors of women of EA contained significantly higher levels of FOXA1[+] cells compared with those of NATs, while this difference was not observed in women of AA (Fig. S8e). These results suggest that the stromal but not the epithelial compartment of the normal breast shows genetic ancestry-dependent variability in cell composition, at least with the markers examined.

## Modeling the effects of PZP cells on breast tumorigenesis

To obtain potential insight into signaling pathway alterations in epithelial and PZP cells as a consequence of their crosstalk, we performed cytokine/chemokine profiling of factors secreted by an immortalized luminal epithelial cell line, a PZP cell line and both co-cultured together (50% of each cell line) for ~24 h. If the expression under the co-culture condition was much higher than expression in either cell type alone, we interpreted those results as showing a cooperative effect on gene expression. If the expression under co-culture conditions was the same or lower than expression in either cell type alone, we interpreted those results as showing no effect of cell-cell interaction. We selected the Luminal-A epithelial cell line KTB34 (EA donor) for this experiment because this cell line upon transformation generates adenocarcinoma, similar to human breast cancer[30,43]. While the luminal epithelial cell line expressed several ligands such as PDGF-AA and osteopontin, which can affect trans-differentiation of PZP cells, PZP cells expressed factors such as EGF, HGF and SDF-1α, which can signal in luminal cells (Fig. 7a; Table S3). Interestingly, IL-6 was produced only under co-culture conditions (Fig. 7a; Table S3). We further confirmed IL-6 production under co-culture conditions at the mRNA level by qRT-PCR (Fig. 7b). We suspect PZP cells produce IL-6 in response to interaction with luminal cells as the basal expression of IL-6 was much higher in PZP cells compared to the luminal cell line (Fig. S9a), and luminal cells secreted IL-1α, which we have previously shown to induce IL-6 in stromal cells[44]. There appears to be specificity in cytokine production under co-culture conditions as we did not observe an elevated production of IL-8 under co-culture conditions of PZP and epithelial cells (Fig. S9b).

## PZP-epithelial cell interaction alters the expression of specific genes

We performed a literature search to identify potential genes whose expression could be altered due to stromal-epithelial cell interaction. For example, transgelin (TAGLN) is known to be a specific marker of smooth muscle differentiation and is highly expressed in the myoepithelial cells and fibroblastic cells of benign breast tissue with limited expression in luminal cells[45,46]. A population of subepithelial cells that lines the entire villus-crypt axis of the intestine expresses high levels of

PDGFRα, Delta Like Canonical Notch Ligand 1 (DLL1), F3, and EGF-family ligand Neuregulin 1 (NRG1)[47]. In addition, NRG1 is also expressed in mesenchymal cells adjacent to the proliferative crypts[47]. Expression of these genes were examined in individual cell types and under co-culture. Three PZP cell lines (KTB32, KTB40, KTB42) and two epithelial cell lines (KTB34, KTB39), as well as co-culture of PZP and epithelial cell lines (50% of each cell line) were used. We observed abundant TAGLN expression in PZP cells, while epithelial cells expressed it at low levels. The expression of TAGLN was further increased under co-culture condition (Fig. 7c). We found a low level of DLL1, F3, and NRG1 expression in PZP cell lines, except for a high level of NRG1 in KTB42. DLL1, F3 and NRG1 are expressed mostly in epithelial cell lines. In co-cultured cells, expression of DLL1 and F3 was additive depending on the cell type (Fig. S9c–e). Taken together, these results indicate that PZP cells correspond to stromal cells that interact with epithelial cells to alter gene expression in a reciprocal manner. However, these cells are unlikely to function similar to subepithelial mesenchymal cells described in the intestine[47].

## Potential role of PZP cells in immune cell modulation in the microenvironment

Several other factors aside from IL-6 have been shown to alter the immune microenvironment. For example, WNT1-inducible-signaling pathway protein 1 (WISP1) expression affects the clinical prognosis by promoting macrophage M2 polarization and immune cell infiltration in pan-cancer and helps to maintain CSC properties in glioblastoma[48]. PZP cell lines displayed higher expression of WISP1 compared to luminal cell lines, while additive expression was observed under co-culture conditions (Fig. 7d). Tenascin-C (TNC) promotes an inflammatory response by inducing the expression of multiple proinflammatory factors in innate immune cells such as microglia and macrophages. TNC drives macrophage differentiation and polarization predominantly toward an M1-like phenotype[49]. TNC is expressed mostly by epithelial cells and this expression was unaffected under co-culture conditions (Fig. 7e). Colony stimulating factor 1 (CSF1) controls both the differentiation and immune regulatory function of macrophages[50]. PZP cell lines displayed higher expression of CSF1 compared to epithelial cell lines and the expression was unaffected under co-culture conditions (Fig. 7f). Osteopontin (SPP1 or OPN), secreted by myofibroblasts, promotes M2 macrophage polarization through the STAT3/PPARγ pathway[51]. PZP cell lines displayed high expression of SPP1 compared to epithelial cell lines and the expression remained additive under co-culture conditions (Fig. 7g). IL-33 is known to be upregulated in metastases-associated fibroblasts, and the upregulation of IL-33 activates type 2 inflammation in the metastatic microenvironment and facilitates eosinophil, neutrophil, and inflammatory monocyte recruitment to lung metastases[52]. Co-culturing of PZP and epithelial cells did not affect IL-33 expression (Fig. 7h). CMTM6 maintains the expression of PD-L1 in tumor cells to regulate anti-tumor immunity[53]. Epithelial cells but not PZP cells expressed higher levels of CMTM6, which was unaffected by co-culture conditions (Fig. S9f). Macrophage migration inhibitory factor (MIF) is an essential cytokine that is involved in the regulation of macrophage function in host defense through the suppression of anti-inflammatory effects of glucocorticoids[54]. Both PZP and epithelial cell lines

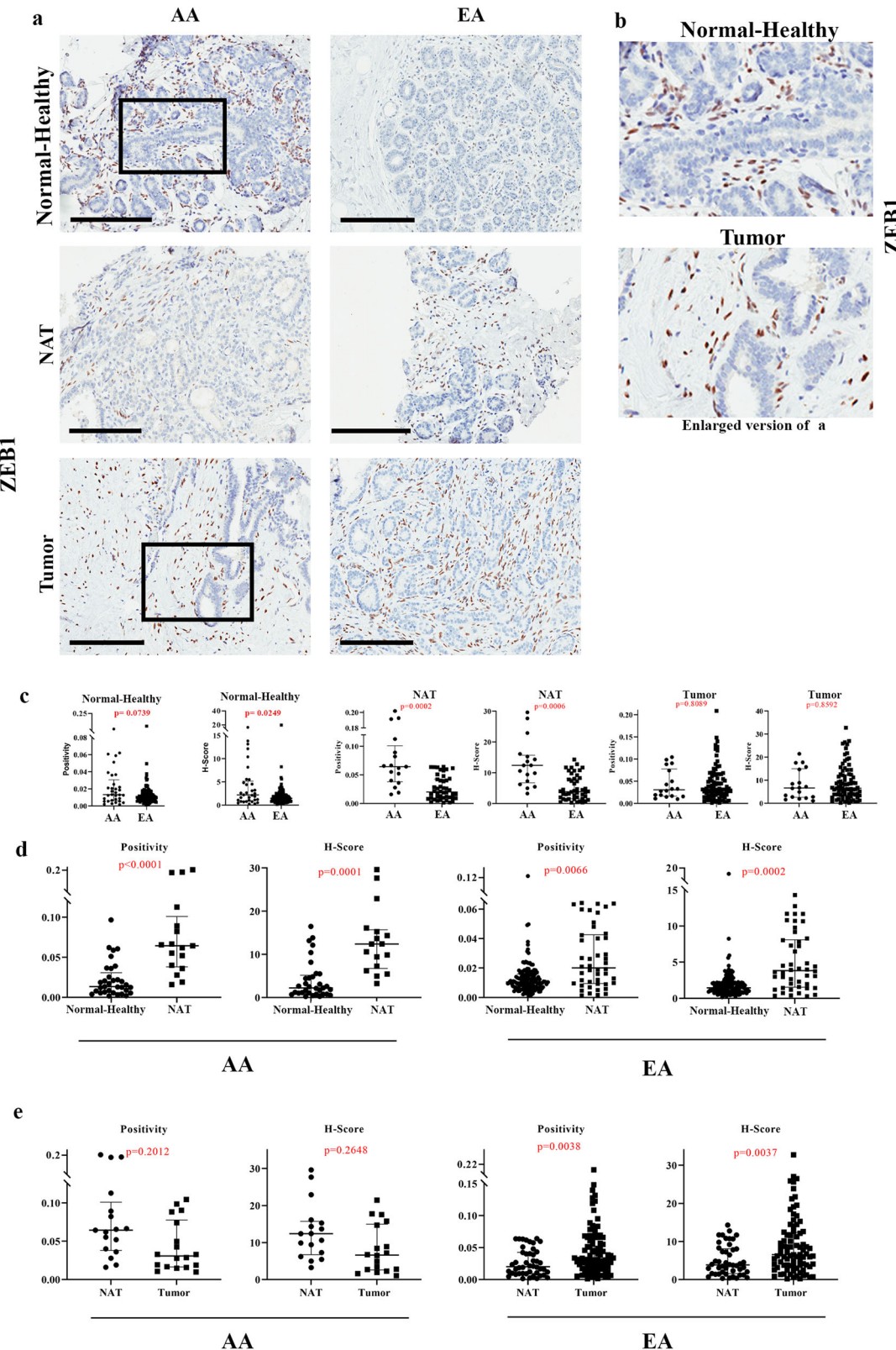

**Fig. 5 | ZEB1 expression pattern in Normal-Healthy, NATs, and breast tumors.**
**a** Representative IHC of ZEB1 in Normal-Healthy, NATs, and/or tumors of women of AA and EA. **b** Enlarged view of ZEB1 expression in Normal-Healthy and tumor. Note ZEB1⁺ cells surround epithelial cell clusters. **c** Differences in ZEB1 expression (positivity and H-score) between Normal-Healthy tissues of women of AA and EA. Differences in ZEB1 expression (positivity and H-score) among Normal-Healthy women of AA and EA analyzed using two-sided Wilcoxon test. **d** Differences in ZEB1 expression (positivity and H-score) between Normal-Healthy and NATs of women of AA (positivity *$p$ = 0.000058998 and H-score *$p$ = 0.000112101) and EA. The data were analyzed using two-sided Wilcoxon test. **e** Differences in ZEB1 expression (positivity and H-score) between NATs and tumors of women of AA and EA analyzed using two-sided Wilcoxon test. (Normal-Healthy-AA ($n$ = 33), EA ($n$ = 144); NAT -AA ($n$ = 12) and EA ($n$ = 31); Tumor -AA ($n$ = 13) and EA ($n$ = 53)). All the data points are shown as mean ± SD. Source data are provided as a Source Data file.

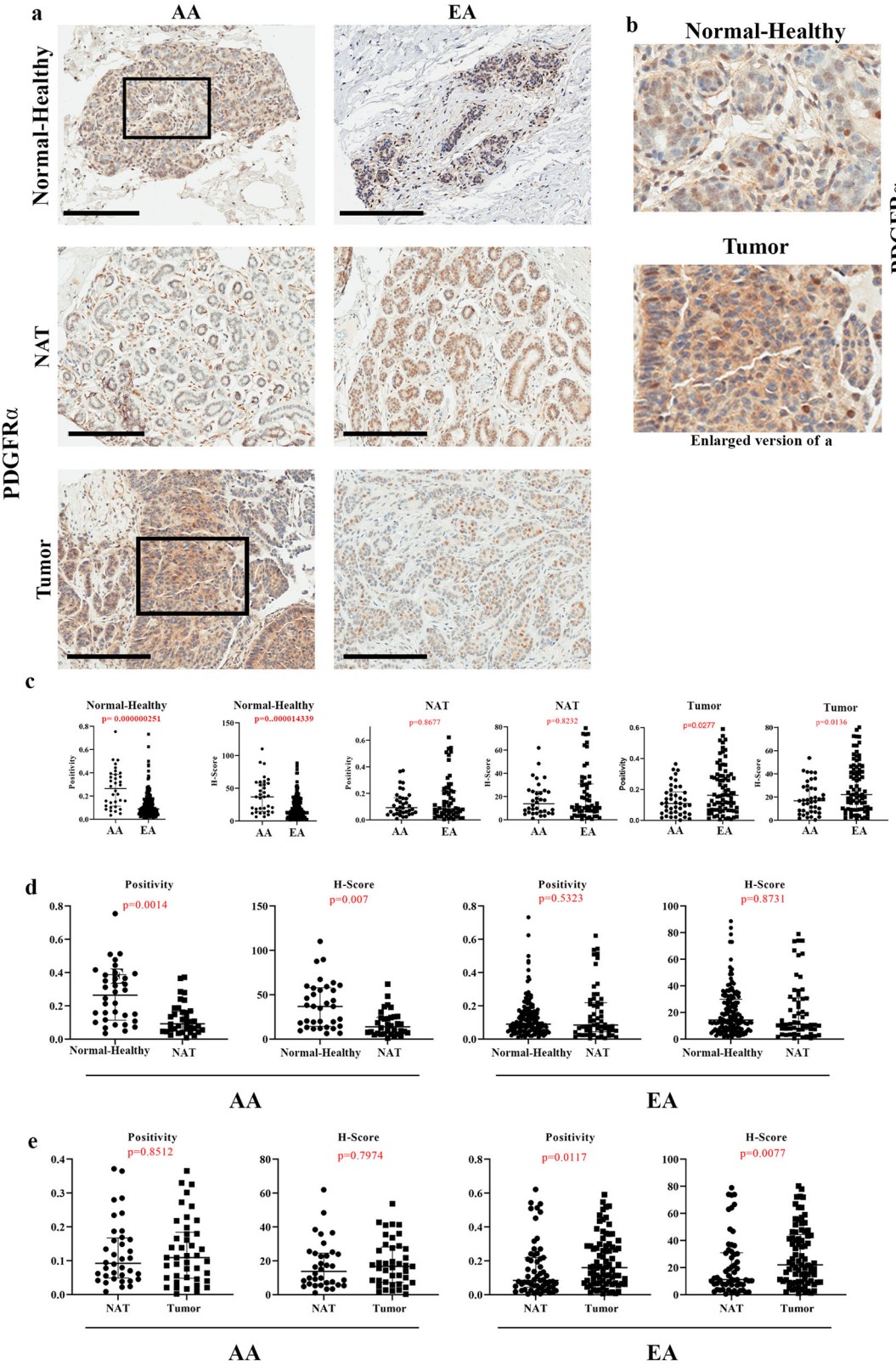

**Fig. 6 | PDGFRα expression pattern in Normal-Healthy, NATs, and breast tumors. a** Representative IHC of PDGFRα in Normal-Healthy, NATs, and/or tumors of women of AA and EA. **b** Enlarged view of PDGFRα expression in Normal-Healthy and tumor. **c** Differences in PDGFRα expression (positivity and H-score) between Normal-Healthy tissues of women of AA and EA analyzed using two-sided Wilcoxon test (positivity *p = 0.000000251 and H-score *p = 0.000014339). **d** Differences in PDGFRα expression (positivity and H-score) between Normal-Healthy and NATs of women of AA and EA analyzed using two-sided Wilcoxon test. **e** Differences in PDGFRα expression (positivity and H-score) between NATs and tumors of women of AA and EA. The data were analyzed using two-sided Wilcoxon test. (Normal-Healthy-AA (*n* = 35), EA (*n* = 154),; NAT -AA (*n* = 24) and EA (*n* = 42); Tumor -AA (*n* = 29) and EA (*n* = 52).) All the data points are shown as mean ± SD. Source data are provided as a Source Data file.

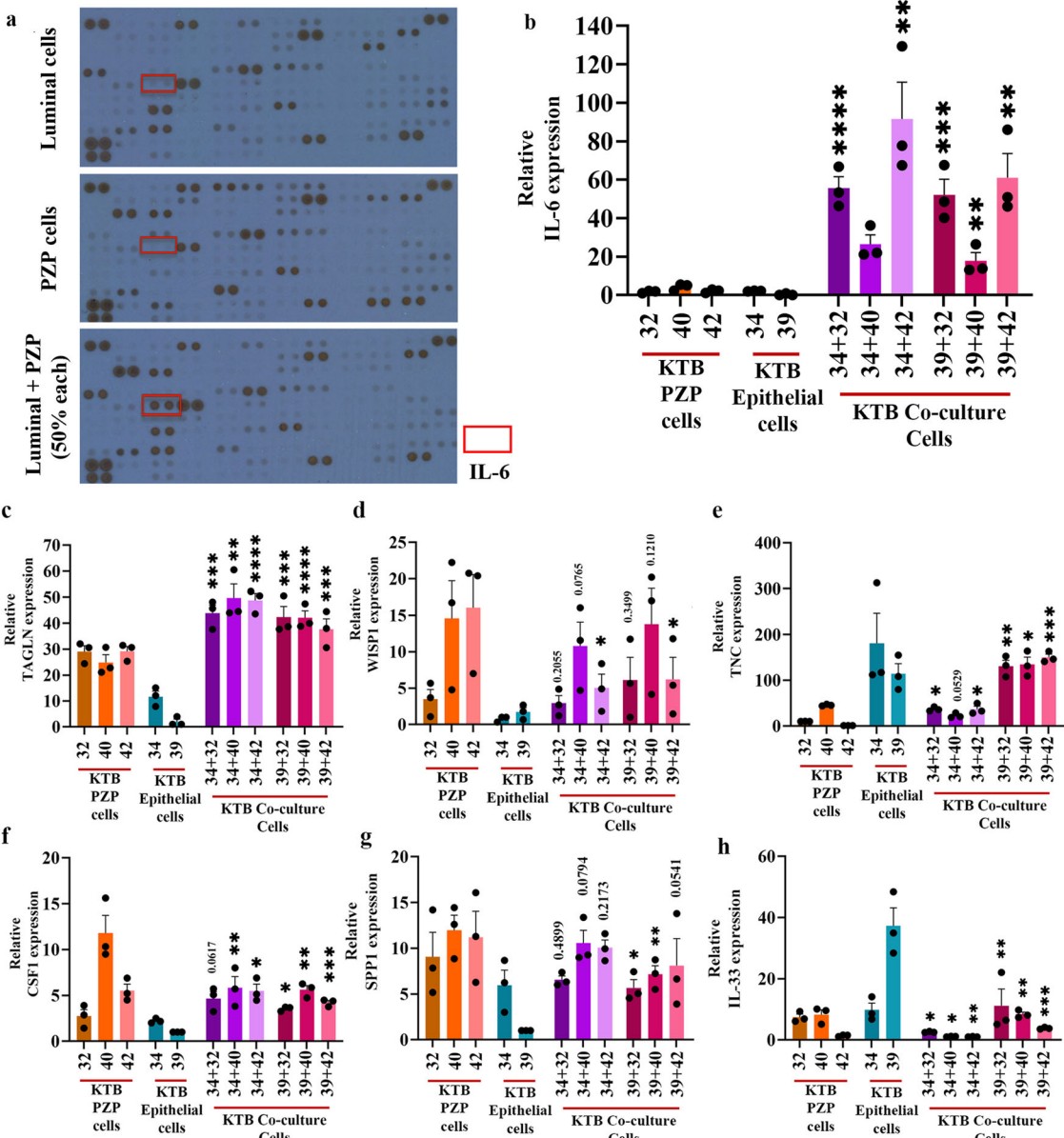

**Fig. 7 | PZP-Epithelial cells interaction alters gene expression. a** IL-6 expression was detected only when luminal and PZP cells were co-cultured. R&D systems cytokine/chemokine array was used to identify secreted factors by epithelial and PZP cells either alone or together (50% each). Expression of genes in PZP (KTB32, KTB40, KTB42), epithelial (KTB34, KTB39), and co-culture of PZP and epithelial cell lines. **b** IL-6 ($n = 3$), KTB34 + 32*$p < 0.0001$, KTB34 + 40*$p = 0.0018$; KTB34 + 42*$p = 0.0018$; KTB39 + 32*$p = 0.0003$; KTB39 + 40*$p = 0.007$; KTB39 + 42*$p = 0.0016$. **c** TAGLN ($n = 3$), KTB34 + 32*$p = 0.0004$; KTB34 + 40*$p = 0.0011$; KTB34 + 42*$p < 0.0001$; KTB39 + 32*$p = 0.0001$; KTB39 + 40*$p < 0.0001$; KTB39 + 42*$p = 0.0002$. **d** WISP1 ($n = 3$), KTB34 + 42*$p = 0.0218$; KTB39 + 42*$p = 0.0456$. **e** TNC ($n = 3$), KTB34 + 32*$p = 0.0401$; KTB34 + 42*$p = 0.0353$; KTB39 + 32*$p = 0.0023$; KTB39 +

40*$p = 0.0158$; KTB39 + 42*$p = 0.0005$. **f** CSF1 ($n = 3$), KTB34 + 40*$p = 0.0059$; KTB34 + 42*$p = 0.0112$; KTB39 + 32*$p = 0.0135$; KTB39 + 40*$p = 0.0016$; KTB39 + 42*$p = 0.0008$. **g** SPP1 ($n = 3$), KTB39 + 32*$p = 0.0348$; KTB39 + 40*$p = 0012$. **h** IL-33 ($n = 3$), KTB34 + 32*$p = 0.0269$, KTB34 + 40*$p = 0.0164$; KTB34 + 42*$p = 0.0045$; KTB39 + 32*$p = 0.0078$; KTB39 + 40*$p = 0.0017$; KTB39 + 42*$p = 0.0005$. Statistical significance ($p$ values) was determined by comparing KTB32/40/42 (PZP cells), KTB34/KTB39 (epithelial cells) and respective co-cultured PZP + epithelial cells as indicated in the figure. *$p < 0.05$, **$p < 0.01$, ***$p < 0.001$, ****$p < 0.0001$ by One-way ANOVA. All the data points are shown as mean ± SEM. Source data are provided as a Source Data file.

expressed MIF, but co-culture conditions did not further affect expression (Fig. S9g). Secretion of MFGE8 can reprogram macrophages from an M1 (proinflammatory) to an M2 (anti-inflammatory but pro-tumorigenic) phenotype[55]. MFGE8 also induces the production of basic fibroblast growth factor, which is responsible for fibroblast migration and proliferation[56]. PZP cell lines displayed high expression of MFGE8 which was unaffected by co-culture conditions (Fig. S9h). Periostin (POSTN) is predominantly secreted by stromal fibroblasts to promote the proliferation of tumor cells. POSTN is also an essential factor for macrophage recruitment in the tumor microenvironment

and is involved in the interactions between macrophages and cancer cells[57]. PZP cell lines displayed high expression of POSTN, and the expression was unaffected by co-culture conditions (Fig. S9i). We noted variation in the expression of select genes amongst PZP cell lines (WISP1, CSF1, NRG1, and POSTN, for example), suggesting inter-individual variability in gene expression in stromal cells, similar to epithelial cell types we described previously[30] and consistent with phenotypic heterogeneity displayed by PZP cell lines (Fig. 3d, for example). Collectively, these results suggest that PZP-epithelial cell interaction in the breast could impact the levels of select chemokines/

cytokines in the breast environment (IL-6 and TAGLN, for example) with consequential effects on the normal and/or tumor immune environment.

### The effects of breast epithelial: PZP cell interaction on trans-differentiation of epithelial luminal progenitor cells

In order to further investigate the intercellular communication between PZP and epithelial cells, we generated stable tdTomato-labeled KTB40 and KTB42 cell lines using pCDH-EF1-Luc2-P2A-tdTomato lentivirus (Fig. S10a). To determine whether co-culturing alters the phenotype of epithelial cells, tdTomato-labeled PZP (KTB40, KTB42), epithelial (KTB34, KTB39), and co-cultured PZP and epithelial cells were analyzed by flow cytometry using CD49f and EpCAM, which can differentiate luminal mature (CD49f$^-$/EpCAM$^+$), luminal progenitor (CD49F$^+$/EpCAM$^+$) and basal (CD49f$^+$/EpCAM$^-$) cells. Co-cultured epithelial cells displayed an increase in CD49f$^+$/EpCAM$^{-/low}$ subpopulation of cells. Reduced CD49f$^{-/low}$/EpCAM$^{+/high}$ cell subpopulations were observed in co-culture conditions (Fig. 8a–c). Isotype controls are shown in Fig. S10b. The tdTomato-Red+ PZP cells on their own were CD49f$^-$/EpCAM$^-$; a small fraction of these cells acquired a CD49f$^+$/EpCAM$^+$ phenotype under co-culture conditions (Fig. 8d, e). These results suggest that similar to mouse mammary fibroadipogenic cells[25], PZP cells can potentially acquire epithelial characteristics under specific conditions. We could not successfully characterize flow sorted CD49f$^+$/EpCAM$^+$ trans-differentiated PZP cells because of dominant growth of a few contaminating PZP cells in the sorted population of cells.

We further confirmed PZP cell mediated transition from luminal to basal characteristics of epithelial cells by ALDEFLUOR assay. Epithelial cells co-cultured with PZP cells lost ALDH-positivity (Fig. S10c). Interestingly, consistent with PZP cells acquiring epithelial characteristics upon co-culture with epithelial cells (based on CD49f$^+$/EpCAM$^+$ positivity noted above), a few of the PZP cells became ALDH$^+$ under co-culture conditions (Fig. S10d). These results suggest that PZP-epithelial cell interactions lead to trans-differentiation of both cell types.

### IL-6 plays a partial role in PZP cell-induced trans-differentiation of epithelial cells

Because co-culturing of PZP cells with epithelial cells caused increased expression of IL-6 in PZP cells, we next examined whether IL-6 plays any role in trans-differentiation of epithelial cells when in contact with PZP cells. The ability of the luminal-like KTB34 cell line to trans-differentiate into CD49f$^+$/EpCAM$^-$ basal like cells when co-cultured with PZP cells was significantly reduced when co-culture conditions contained neutralizing antibody against IL-6R (Fig. S11a, b). The effect of IL-6R antibody in reducing trans-differentiation of KTB39 was limited. Unlike KTB34 cells, which have luminal features, KTB39 from a woman of AA displays normal-like intrinsic subtype features[30] and is possibly susceptible to trans-differentiation by other cytokines/chemokines.

We evaluated the effect of IL-6 inhibition on CD44, EpCAM, CD24 and CD10 cell surface marker profiles of KTB34 and KTB39 cells with and without co-culturing with KTB40 or KTB42 PZP cell lines. While co-culturing increased the percentage of CD44$^+$/EpCAM$^{low}$ KTB34 and KTB39 cells, IL-6R neutralizing antibody had minimum effect on this trans-differentiation (Fig. S11c, d). Similarly, IL-6R antibody treatment did not influence the effects of PZP cells in epithelial cells losing CD10 expression (gain of CD24$^+$/CD10$^{low}$ phenotype) (Fig. S12a, b).

We next evaluated the effect of IL-6 inhibition on the STAT3 signaling pathway and measured the phosphorylation levels of STAT3 (Y705 and S727) in KTB34 cells treated with conditioned media (CM) from KTB34, KTB40, and co-cultured KTB34 and KTB40 cells. These two phosphorylation were selected because they differentially regulate epithelial-mesenchymal-transition (EMT) to mesenchymal-epithelial-transition (MET) switch during metastasis[58]. CM from KTB34

and KTB40 cell lines induced phosphorylation of STAT3 (Y705) at variable levels but the induction was highest when CM from co-cultured cells was used (Fig. 8f). IL-6R antibody drastically reduced KTB40 and co-culture CM-mediated pSTAT3 (S705) phosphorylation. Because of very high levels of basal phosphorylation at S727 residue due to culture conditions, the effect of PZP cell CM on S727 phosphorylation was modest. These results suggest that enhanced IL-6 production by PZP cells as a consequence of their interaction with epithelial cell leads to STAT3 activation in epithelial cells.

### Normal breast and/or breast tumors of women of African ancestry contain higher levels of phospho-STAT3

Since PZP-epithelial cell interaction resulted in elevated IL-6 expression in vitro and PZP cells are present at higher levels in the normal breast of women of AA, it is expected that signals downstream of IL-6 should be higher in the breast tissues of women of AA compared to women of EA. We used phospho-STAT3 as a surrogate marker to determine IL-6 activity in the normal breast tissues. Antibody against S727 but not S705 was sensitive in immunohistochemistry. Although cells in the Normal-Healthy breast tissues contained lower levels of phospho-STAT3 as we could estimate only positivity, positivity was still higher with healthy breast tissues of women of AA compared to women of EA (Fig. 9a–c). Modestly higher levels of phospho-STAT3 were also observed in tumors of women of AA compared to women of EA (Fig. 9c). Consistent with elevated levels of PZP cells in NATs compared to Normal-Healthy in women of EA, phospho-STAT3 levels were higher in NATs compared to Normal-Healthy of women of EA (Fig. 9d). We analyzed several publicly available proteomics and transcriptomics datasets[59,60] using the UALCAN database[61] to determine whether STAT3 activity levels, as measured through phosphoprotein levels or STAT3 gene signatures[62], are higher in breast tumors of African American women compared to non-Hispanic White women (note information in these cases is limited to self-reported race and ethnicity). Significantly elevated STAT3 phosphorylation at specific residues of STAT3 as well as STAT3 gene signatures were noted in tumors from African American women compared to tumors from Asian women but not between tumors from African American and non-Hispanic White donors. Overall, our results and these publicly available data support the impact of genetic ancestry on phosphorylation/activity status of STAT3.

### Transformed PZP cells generate metaplastic carcinoma

We recently reported that cell-of-origin but not oncogenic mutations determine the histotypes of breast tumors using breast epithelial cell lines derived from multiple donors and a defined set of oncogenes[43]. We used the same strategy to determine whether PZP cells are cell-of-origin of specific malignancy of the breast. Mutant Ras is one of the potent oncogenes used to transform breast epithelial cells in vitro[63]. Although initially considered not a relevant oncogene in breast cancer and breast cancer-related studies that utilized Ras oncogenes were often dismissed upon by reviewers, recent studies have clearly shown the role of Ras oncogene in endocrine resistance and metastasis of luminal breast cancer[64]. Furthermore, the majority of breast cancer cell lines used as model systems have either mutation in Ras genes themselves or in downstream effectors or negative regulators of Ras pathway. Ras pathway is the major signaling pathway activated in TNBCs and Claudin-low breast cancer subtype[65]. We transformed PZP cell lines with HRas$^{G12V}$, SV40-T/t antigen and the combination of both HRas$^{G12V}$ and SV40-T/t antigens using lentivirus, since this combination was the most effective in breast epithelial cell transformation[43]. SV40-T/t antigens impair tumor suppressor functions of retinoblastoma and p53 proteins, inhibit the activity of protein phosphatase 2 A and generate gene expression signatures found in aggressive breast cancers[66]. Western blotting was used to detect the overexpression of mutant HRas$^{G12V}$ (Fig. S13a), SV40-T/t antigens (Fig. S13b), and combination of

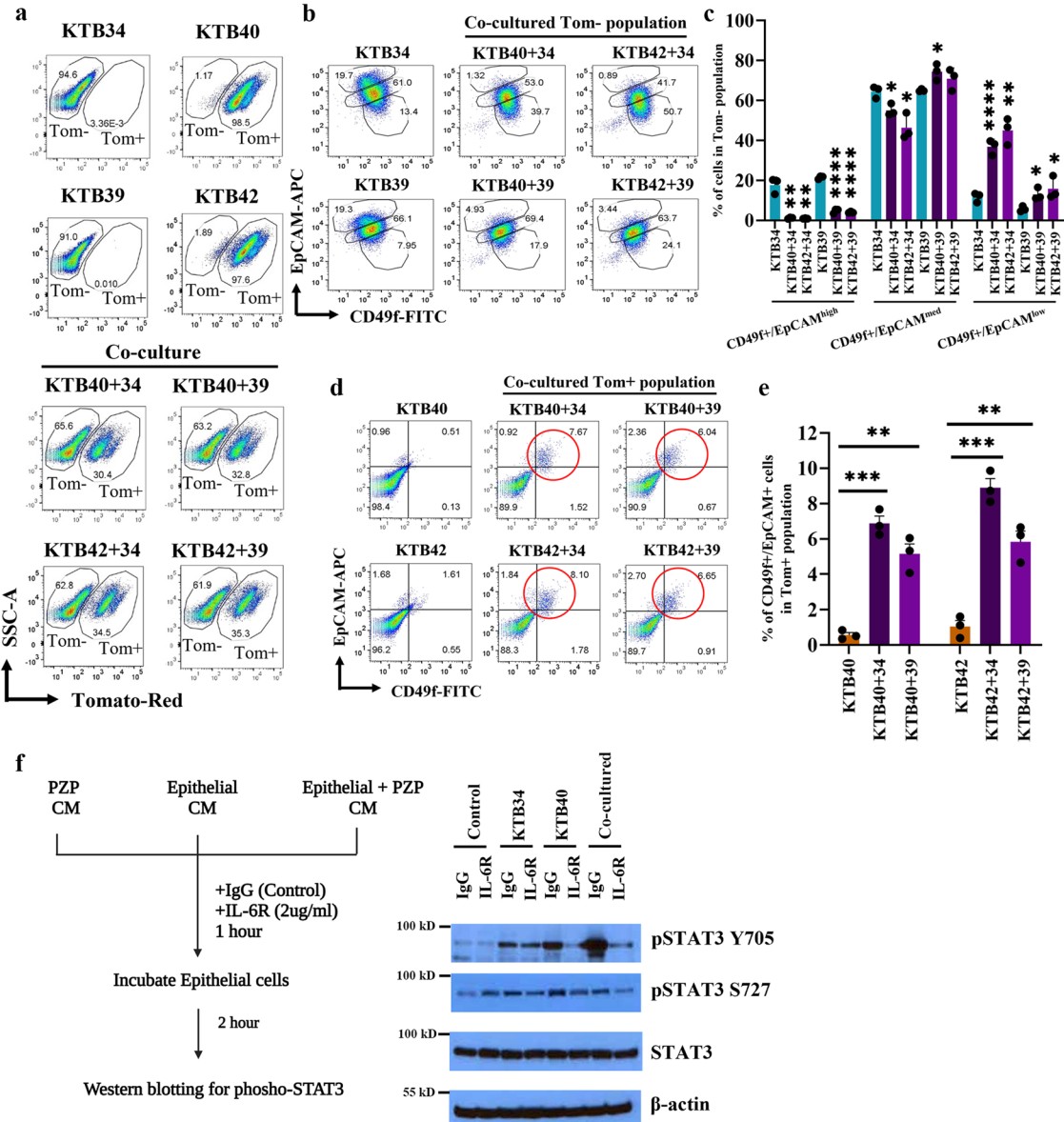

**Fig. 8 | The effects of PZP cells on trans-differentiation of epithelial cell lines.**
**a** Gating of tomato- (Tom⁻) and tomato+ (Tom⁺) populations of KTB34, KTB39, and co-cultured KTB40/KTB42 and KTB34/KTB39 cell lines ($n = 3$). PZP cell lines are Tom⁺. **b** CD49f and EpCAM staining patterns of KTB34, KTB39, and co-cultured KTB40/KTB42 and KTB34/KTB39 cell lines ($n = 3$). Isotype controls are shown in Fig. S10b. **c** Quantification of CD49f⁺/EpCAM^high, (KTB40 + 34*$p = 0.002$; KTB42 + 34*$p = 0.0019$; KTB40 + 39*$p < 0.0001$; KTB42 + 39*$p < 0.0001$), CD49f⁺/EpCAM^med (KTB40 + 34*$p = 0.0186$; KTB42 + 34*$p = 0.0124$; KTB40 + 39*$p = 0.0183$), and CD49f⁺/EpCAM^low (KTB40 + 34*$p = 0.0006$; KTB42 + 34*$p = 0.0012$; KTB40 + 39* $p = 0.0185$; KTB42 + 39*$p = 0.0414$) populations in Tom− cell population ($n = 3$). **d** CD49f and EpCAM staining patterns of KTB40, KTB42, and co-cultured KTB40/KTB42 and KTB34/KTB39 cell lines. Only Tom⁺ population was analyzed ($n = 3$).

**e** Quantification of CD49f⁺/EpCAM⁺ population among Tom⁺ population ($n = 3$). (KTB40 + 34*$p < 0.0001$; KTB40 + 39*$p = 0.0015$; KTB42 + 34*$p = 0.0002$; KTB42 + 39*$p = 0.0024$) (**f**) pSTAT3 Y705, pSTAT3 S727, and STAT3 expression levels in KTB34 cells treated with CM obtained from KTB34, KTB40, and co-cultured KTB34 and KTB40 cells followed by IgG control and IL-6R neutralizing antibody treatment ($n = 3$). CM was treated with IgG or IL-6R neutralizing antibodies for 1 h and CM treatment lasted for 2 h. β-actin was used as an internal control ($n = 3$). In this representative experiment, the same batch of extracts was used to run four blots and each blot was probed with indicated antibodies separately. Statistical significance derived (*$p < 0.05$, **$p < 0.01$, ***$p < 0.001$, ****$p < 0.0001$) using Two-tailed $t$-test. All the data points are shown as mean ± SEM. Source data are provided as a Source Data file.

both HRas^G12V and SV40-T/t antigens (Fig. S13c). Phase contrast images of PZP transformed cell lines are shown in Fig. S13d. Transformation of PZP cells with activated HRas^G12V increased the fraction of cells that have acquired epithelial phenotype and express EpCAM, particularly in KTB42 (Fig. 10a, b; Fig. S13d). Transformed PZP cells expressed the basal cell marker CD49f with inter-individual variability (Fig. 10a; Fig. S14a for isotype control). Transformation also altered the cell surface profiles of the mesenchymal stem cell marker CD90 with a few of PZP lines demonstrating loss of CD90 expression (Fig. 10c; Fig. S14a). Transformed PZP cells were PROCR⁺ and CD44⁺ (Fig. 10b

Fig. S14b). Thus, PZP cells further undergo trans-differentiation upon transformation by acquiring CD49f expression.

We next determined whether cells expressing oncogenes are tumorigenic in NSG mice. Indeed, five million transformed cells in 50% matrigel implanted into the mammary gland of 6–7-week-old female NSG (NOD/SCID/IL2Rgnull) mice ($n = 5$) progressed into tumors. All animals injected with HRas^G12V + SV40-T/t and four out of five animals injected with HRas^G12V transformed cells generated tumors. Parental immortalized cell lines ($n = 5$) did not generate tumors. Tumor was resected and analyzed by H&E staining and expression of luminal

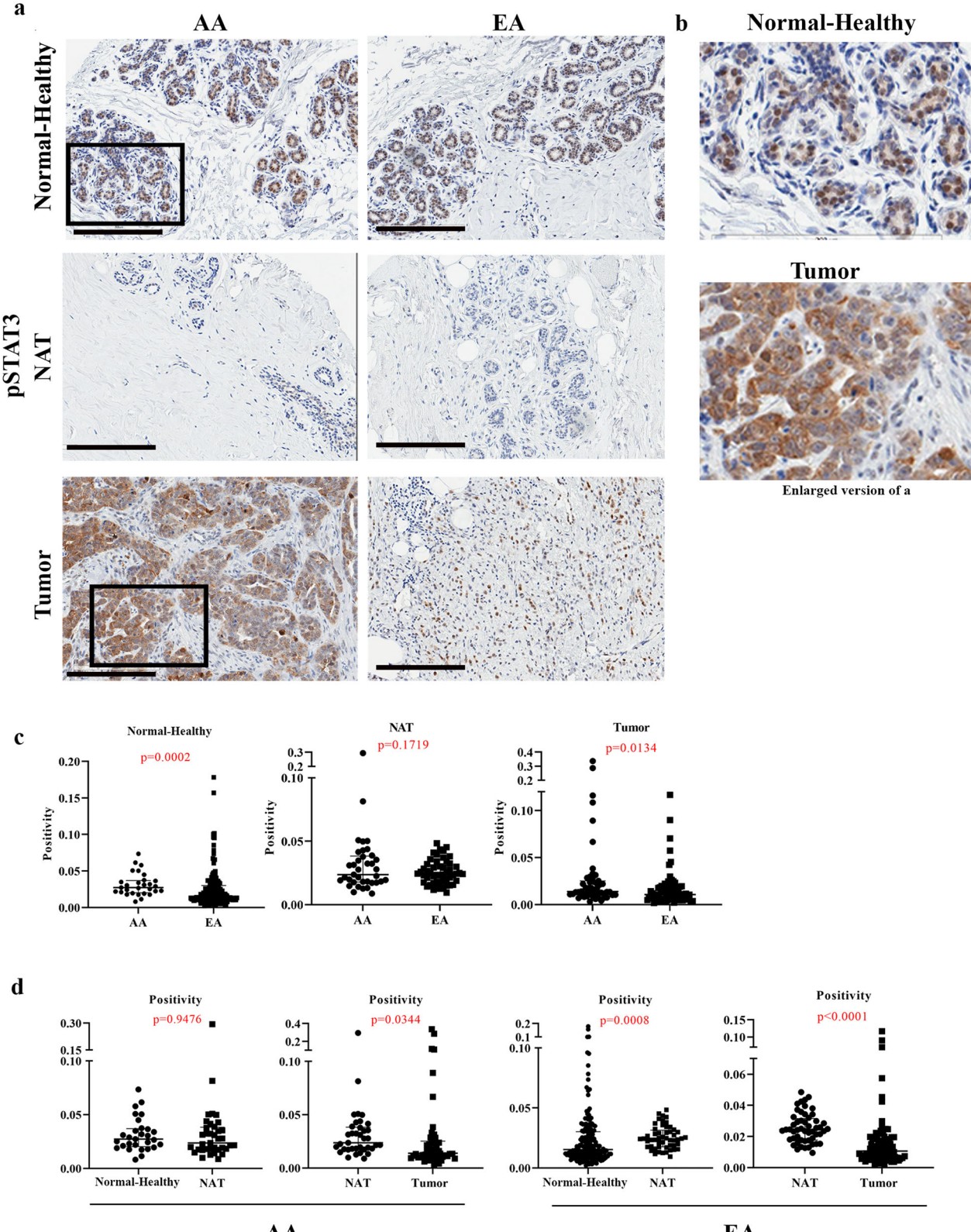

**Fig. 9 | Phospho-STAT3 levels in Normal-Healthy, NATs, and breast tumors.**
**a** Representative IHC of pSTAT3 in Normal-Healthy, NATs, and/or tumors of women of AA and EA. **b** Enlarged view of pSTAT3 expression in Normal-Healthy and tumor. **c** Differences in pSTAT3 positivity between Normal-Healthy, NATs and tumors of women of AA and EA. **d** Differences in pSTAT3 positivity between Normal-Healthy and NATs and between NATs and tumors of women of AA and EA analyzed using two-sided Wilcoxon test. (Normal-Healthy, AA = 23, EA = 164); NAT -AA ($n$ = 24), EA ($n$ = 31); Tumor -AA ($n$ = 32), EA ($n$ = 41).) All the data points are shown as mean ± SD. Source data are provided as a Source Data file.

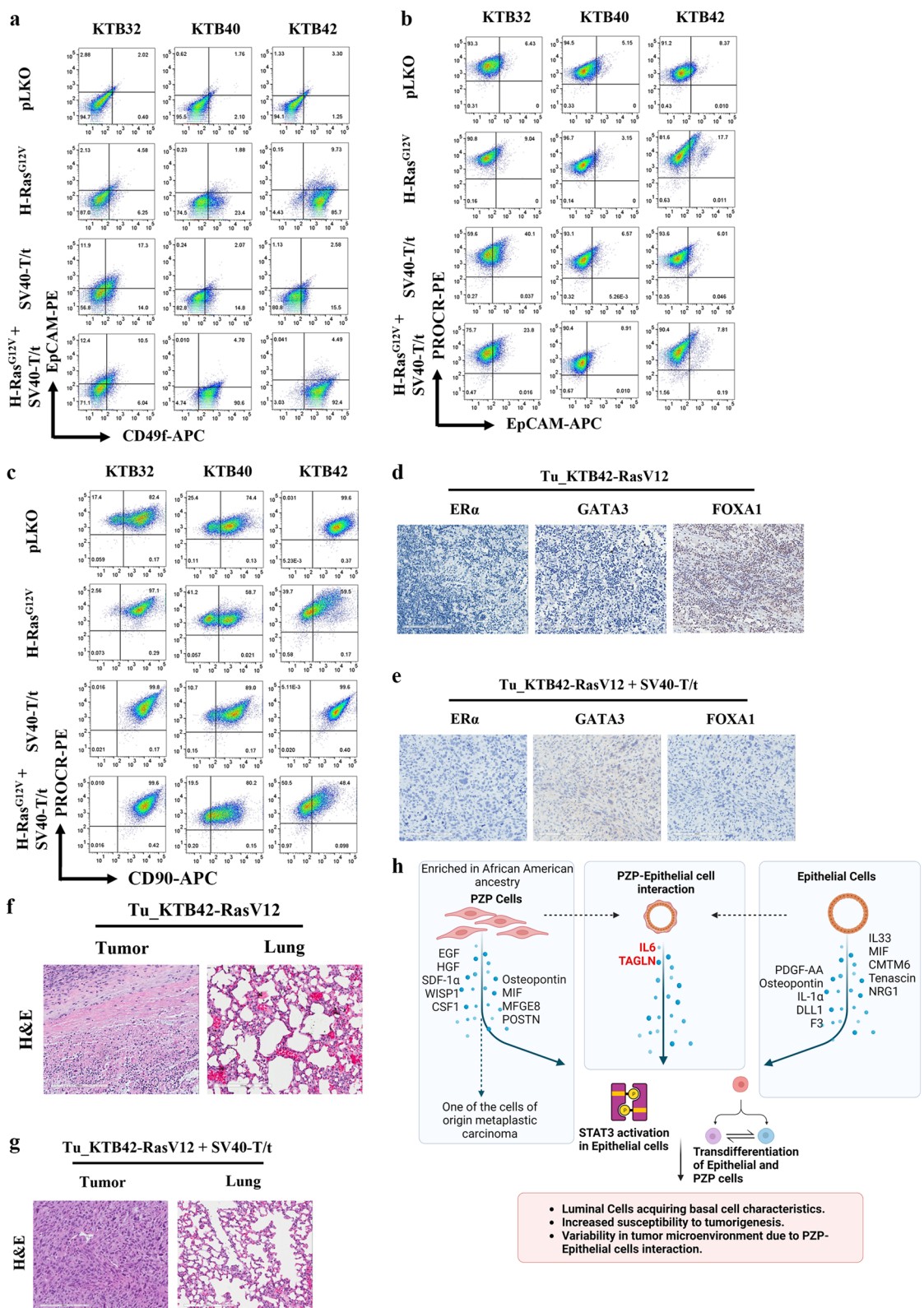

**Fig. 10 | PZP cells transformed with Ras and SV40-T/t antigens are tumorigenic.**
**a** CD49f and EpCAM staining patterns of immortalized and transformed PZP (KTB32, KTB40 and KTB42) cell lines. CD49f⁺/EpCAM⁻, CD49f⁺/EpCAM⁺ and CD49f⁻/EpCAM⁺ cells correspond to stem/basal, progenitor and differentiated cells, respectively ($n = 3$). **b** PROCR and EpCAM staining patterns of immortalized and transformed PZP (KTB32, KTB40, and KTB42) cell lines ($n = 3$). **c** CD90 and PROCR staining patterns of immortalized and transformed PZP (KTB32, KTB40, and KTB42) cell lines ($n = 3$). **d** IHC analyses of luminal markers ERα, GATA3 and FOXA1

in tumors developed from KTB42-HRas$^{G12V}$ transformed cells ($n = 5$). **e** IHC analyses of luminal markers ERα, GATA3 and FOXA1 in tumors developed from the KTB42-HRas$^{G12V}$ + SV40-T/t antigen transformed cells ($n = 5$). **f** Tumor developed from KTB42-HRas$^{G12V}$ transformed cells did not show lung metastasis ($n = 5$). **g** Tumor developed from KTB42-HRas$^{G12V}$ + SV40-T/t antigen transformed cells did not show lung metastasis ($n = 5$). H&E staining shows the type of tumors. **h** Schematic view of study findings.

markers estrogen receptor alpha (ERα), GATA3, and FOXA1 and cytokeratins CK5/6, CK8, CK14 and CK19 was examined. The luminal cells express cytokeratin 19 (CK19), while basal cell types express cytokeratin 5/6 (CK5/6) and cytokeratin 14 (CK14), and cells expressing both CK14 and CK19 show luminal progenitor phenotype or correspond to luminal/basal hybrid cells of the breast[67]. KTB42-HRas[G12V] cell-derived tumor was ERα[-]/GATA3[-]/FOXA1[+] (Fig. 10d). KTB42-HRas[G12V]-derived tumor was also CK5/6[-]/CK8[-]/CK14[-]/CK19[-] (Fig. S14c). KTB42 cell line transformed with both mutant HRas[G12V] and SV40-T/t antigen also developed tumors in NSG mice. KTB42- HRas[G12V] + SV40-T/t cell-derived tumor was ERα[-]/GATA3[-]/FOXA1[-] (Fig. 10e), and CK8[-]/CK14[-]/ CK19[-] (Fig. S14d). Unlike luminal breast epithelial cell-derived tumors obtained after transformation with the same set of oncogenes[43], which metastasized to lungs, these tumors did not show extensive lung metastasis (Fig. 10f, g). Histologically, these tumors were metaplastic carcinomas, which comprise 0.08–0.2% of all breast neoplasms (Fig. 10g)[29].

To further establish relationship between PZP cells and metaplastic carcinomas, we compared characteristics of PZP cells with published metaplastic carcinoma protein/gene expression datasets[68,69]. We had previously reported RNA-seq data of immortalized PZP cell lines KTB40 and KTB42 and six immortalized breast epithelial cell lines (KTB21, 22, 26, 34, 37, and 39)[30]. Similar to metaplastic carcinomas that show EMT characteristics with enhanced mesenchymal gene expression and claudin low phenotype, PZP cells expressed higher levels of EMT-associated genes ZEB1, ZEB2, SNAI1 and TWIST1 but lower levels of E-Cadherin and Claudin-7 (CLDN7) compared to epithelial cells. PZP cells did not express CLDN3 and CLDN4. Gene Set Enrichment Analysis (GSEA) of PZP cells compared to epithelial cells revealed enrichment of EMT, IFNγ and apical junction signatures, and reduced OXPHOS signatures (Fig. S14E), similar to metaplastic carcinomas[68]. Since cell-of-origin gene signatures instead of mutation driven signatures predominate in 33 cancer types that have been examined[70], these results further support the possibility of PZP cells being one of the cells-of-origin of metaplastic carcinomas.

To ensure that tumors originate from PROCR[+]/EpCAM[-] cells instead of a minor fraction of contaminating epithelial cells, we sorted PROCR[+]/EpCAM[-] cells from the KTB42 cell line (Fig. S15a). Gate for the sort was placed in a such a way that there is limited possibility of EpCAM[+] cell contamination. Sorted PROCR[+] KTB42 cells transformed with mutant HRas[G12V] and SV40-T/t antigen (five million) in 50% matrigel implanted into the mammary gland of 6–7-week-old female NSG mice (n = 6) progressed into tumors. We characterized the cell line established from the tumor to determine the expression of PROCR and other phenotypic markers. Human specific antibody against Na[+]/ K[+] ATPase CD298 (ATP1B3) cell surface marker was used to sort CD298-enriched human tumor cell populations from mouse stromal cells (Fig. S15a)[43]. A cell line established from tumor of KTB42-HRas[G12V] + SV40-T/t transformed cells showed PROCR[+] cells similar to transformed cells in vitro (Fig. S15a). Tumor-derived cell line expressed the basal cell marker CD49f similar to transformed cells. Tumor-derived cell line displayed an increase in CD90[high]/CD73[+] cell population compared to transformed cells in vitro (Compare Fig. S15a with Fig. 10c). Tumor-derived cells were CD44[+] and CD10[+] similar to transformed cells (Fig. S15a). H&E, PROCR, and EpCAM staining of resected tumors revealed PROCR[+] spindle-like tumor cells with pockets of PROCR[+]/EpCAM[+] tumor cells (Fig. S15b, c). No lung metastasis was observed.

To further clarify that tumors are derived from PROCR[+]/EpCAM[-] cells and few of the tumor cells trans-differentiate into EpCAM[+] cells, we generated a single cell-derived clone (D4) from PROCR[+]/EpCAM[-] cell population of KTB42-HRas[G12V] + SV40-T/t transformed cells by sorting PROCR[+]/EpCAM[-] cells using flow cytometry (Fig. S15d). Single cell-derived clone D4 was PROCR[+], but a small fraction of cells transdifferentiated into EpCAM[+] cells (Fig. S15d). Five million single cell derived PROCR[+] KTB42-HRas[G12V] + SV40-T/t transformed clone D4 cells in 50% matrigel were implanted into the mammary gland of 6–7-week-old female NSG mice (n = 5). All animals injected with the single cell derived PROCR[+] KTB42-HRas[G12V] + SV40-T/t transformed clone D4 generated tumors (Fig. S15e, f). Similar to tumors derived from bulk PROCR[+]/EpCAM[-] transformed cells, tumors derived from the single cell derived clone were also PROCR[+] and spindle shaped with pockets of EpCAM[+] tumor cells. Collectively these results suggest that PZP cells can generate tumors similar to metaplastic breast carcinomas with pockets of malignant epithelial cells[71]. A summary of major findings of this study is presented schematically in Fig. 10h (All the elements described in figure are generated through this study).

## Discussion

As per the most recent American Cancer Society estimate, the breast cancer mortality rate in African American women is significantly higher compared to non-Hispanic White women, while breast cancer incidence in African American women is lower[72]. This difference cannot be explained solely by differences in socioeconomic status or access to health care[15]. Additional possibilities for this disparity include the differing biology of the normal breast between these two groups, which may contribute to altered susceptibility to tumor initiation, progression and/or metastasis. This study focused on one particular stromal cell type, which is named as PZP cells, that can contribute to genetic ancestry-dependent variability in breast biology. While PZP cells are intrinsically higher in the normal breasts of women of AA, their numbers increase in women of EA only in breasts with cancer. We speculate that while tumors in women of AA can take advantage of intrinsically higher PZP cells, there is a lag period in carcinogenesis in tissue from women of EA requiring tumor cell: residual PZP cell interaction to promote expansion of PZP cells. We also propose two distinct roles of PZP cells in breast tumorigenesis. First, as a source of IL-6 and TAGLN in the tumor microenvironment, PZP cells promote luminal cells acquiring luminal-basal hybrid phenotype[27], alter immune microenvironment and thus predisposing luminal breast epithelial cell-derived tumors to be intrinsically aggressive. We suspect this is the major contribution of PZP cells in aggressive TNBCs in women of AA. The second is that PZP cells themselves are one of the cells-of-origin of metaplastic cancers.

Cell surface marker profiling and trans-differentiation studies indicate similarities between PZP cells of the breast and tissue resident mesenchymal stem cells, which have trans-differentiating and regenerative capacity[73]. PZP cells contain several subpopulations, CD90[-]/ CD73[high], CD90[low]/CD73[high], CD90[med]/CD73[high] and CD90[high]/CD73[high]. These cells also differentiate into adipocyte and osteogenic lineages. A similar cell type in the mouse mammary gland functions as an adipogenic progenitor and differentiates into luminal and basal epithelial cells during mammary morphogenesis[25]. We observed epithelial differentiation of PZP cells only upon transformation by HRas[G12V] or when co-cultured with epithelial cells. In this respect, PZP cells are the breast counterpart of PDGFRα[+] intestinal stromal progenitors or fibroadipogenic progenitors (FAPs) of the skeletal muscle. FAPs are multipotent, and are capable of adipogenesis, fibrogenesis, osteogenesis and chondrogenesis[47,73]. While intestine and skeletal muscle undergo extensive regeneration upon injury, normal breast undergoes regeneration or modification during the pregnancy-lactation-involution cycle. Therefore, it is not surprising that such stromal cells exist in the breast.

PZP cells also displayed stromal fibroblast-like properties. Morsing M. et al. reported that a distinct fibroblast lineage in the human breast, which supports luminal epithelial progenitor growth and possessing adipogenic capacity, gathers around TDLUs[35]. These cells are likely derived from lobular origin, which is evident by the enrichment of CD105[high]/CD26[low] population. We found that PZP cells stably maintain the CD105[high]/CD26[low] phenotype with extended culture.

Lobular-like fibroblasts regulate epithelial morphogenesis and differentiation typical of the TDLU through TGFβ signaling pathway[34,35]. Early myoepithelial progenitors are susceptible to cues from lobular-like and interlobular-like fibroblasts in terms of their luminal differentiation repertoire[34,35]. Since PZP cells expressed several secretory factors more than luminal cells (Table S3), few of these factors could provide specific cues to luminal cells.

Co-culture experiments provided a clue to how alterations in number of PZP cells in the breast could influence the breast microenvironment (summary in Fig. 10h). For example, we observed a higher level of TAGLN expression under co-culture conditions compared to either cell types cultured separately. Recent studies have shown that TAGLN localizes in the cytoplasm of benign and malignant tumor cells in the breast tissue[45]. It is strongly expressed in the cytoplasm of myoepithelial and fibroblastic cells of the benign breast tissue; however normal luminal cells are predominately negative or have a very weak cytoplasmic expression. TAGLN is differentially expressed among molecular breast cancer subtypes and predominantly upregulated in higher grade TNBCs[45]. Expression of TAGLN is positively correlated with high Ki-67 and low ER and PR status. Other studies showed that TAGLN is directly associated with migratory and invasive ability of tumorigenic cells and TAGLN overexpression increases the invasiveness of both tumorigenic and nontumorigenic cells[74].

PZP cell-epithelial cell interaction leading to elevated local production of IL-6 could also have a profound effect on the microenvironment. In this respect, genetic ancestry-dependent variability in breast tumor immune microenvironment has already been demonstrated, particularly in relation to CD8+ T-cell exhaustion[75] and response to immunotherapy[9]. IL-6 is a pleiotropic cytokine that contributes to EMT and drives metastasis in ER+ breast cancer[76]. IL-6 activates JAK/STAT3 signaling, which drives tumor proliferation, survival, invasiveness, angiogenesis and immunosuppression[77]. Consistent with the possibility of genetic ancestry-dependent variability in PZP cells causing elevated local IL-6 levels and downstream signaling, we observed elevated phospho-STAT3 levels in normal and tumors from women of AA compared to EA and in in vitro studies, neutralizing antibodies against IL-6 receptor reduced PZP cell-mediated activation of STAT3 in epithelial cells.

Co-culturing of epithelial cells with PZP cells caused changes in the phenotype of epithelial cells with a fraction of luminal progenitor cells acquiring basal like properties, which was partially dependent on IL-6. Recent studies have suggested that luminal cells with basal cell properties are enriched for gene signatures of basal-like breast cancers and number of cells with this property increase with aging[27]. Whether locally produced IL-6 is responsible for luminal cells acquiring a basal phenotype remains to be determined although there is evidence in the literature for IL-6 conferring plasticity to epithelial cells[78]. Currently, at least four IL-6 targeting drugs are either approved or in clinical trials[79]. It remains to be determined whether IL-6 targeting drugs are more effective against breast tumors of women of AA compared to women of EA.

Metaplastic carcinoma of the breast, although rare, is more common in African American women compared to non-Hispanic White women[23]. Our results indicate that PZP cells are one of the cells-of-origin of this cancer type. Identifying cell-of-origin allows further evaluation of signaling pathways activated in this cancer type and eventually development of targeted therapies. Metaplastic carcinomas of the breast are CD49f+/EpCAM−[80] and can be subdivided into metaplastic, claudin- (META-CLOW) and metaplastic-squamous, and metaplastic-non-CLOW, non-Squamous. PZP cells acquired CD49f+/EpCAM− phenotype upon transformation and tumors derived from these cells displayed squamous features, suggesting that PZP cells are the cell-of-origin of metaplastic-squamous subtype.

Mechanistically, the cause of PZP cell numbers being elevated in healthy breasts of women of AA is unknown but unlikely related to socioeconomic stress (Fig. S5). FAPs, which are similar to PZP cells, increase in numbers in skeletal muscle of individuals with type II diabetes[81]. Whether type II diabetes, which is more common in African American than non-Hispanic White women[82], is responsible for or related to increased PZPs in the breasts of women of AA remains to be determined.

In summary, this study identified and established PZP cell lines from healthy breasts of women of AA and these cells phenotypically resemble tissue resident mesenchymal stromal/stem cells with trans-differentiation capabilities and can regulate signaling in neighboring epithelial cells. Although we identified only two factors, TAGLN and IL-6, whose levels in the tissue microenvironment could be influenced due to PZP-epithelial cell interaction, there may be additional signaling pathway activation as a consequence of this interaction and this could potentially influence the microenvironment. Because of the ability of PZP cells to influence signaling in epithelial cells and differences in their number based on genetic ancestry, signaling pathway in epithelial cells of the breast may show genetic ancestry-dependent variability under both normal and cancerous conditions and drugs that target crosstalk between these two cell types can be new therapeutic agents. These drugs may show genetic ancestry-dependent variability in their potency. Therefore, in addition to including patients of different genetic ancestry in clinical trials, evaluation of outcomes of the clinical trials need to take genetic ancestry into consideration. A few therapeutics, either as monotherapy or in combination, may be effective only in patients of specific genetic ancestry.

## Methods

### Primary cell culture, immortalization and lentiviral transduction

PZP cell lines were created from fresh or cryopreserved, de-identified normal breast tissues donated to KTB by healthy self-reported African American women; all subjects provided written informed consent prior to donation. African ancestry of the donors was subsequently verified via analysis of ancestry-informative markers. All experiments were carried out in accordance with the approved guidelines of the Indiana University Institutional Review Board. International Ethical Guidelines for Biomedical Research involving human subjects were followed. Generation of primary cells from cryopreserved tissues, culturing method and media composition have been described previously[83]. Briefly, cryopreserved tissues were thawed, minced, and incubated in collagenase/hyaluronidase mix (300 µl in 2.7 ml of media, #07919, Stem Cell Technologies) plus ROCK inhibitor (5 µM, #ALX-270-33-005, Enzo Life Sciences) for 2 h at 37 C. Debris were removed by filtering through 70 micron filter. Cells were pelleted by centrifugation (150 g for 5 min), washed, and plated on 60 mm dishes pre-coated with conditioned media from 804 G cell line. While primary breast tissues cultured from self-reported non-Hispanic White women generated predominantly tightly attached cuboidal epithelial cell colonies, tissues from African American donors yielded both loosely attached elongated cell clusters and cuboidal epithelial colonies. Differential trypsinization was used to separate two population of cells for immortalization. Cells were trypsinized for 2 min at room temperature, which allowed loosely attached elongated cells to peel off the plates and those cells were collected and replated as PZP cells. Cells were immortalized by human telomerase gene (hTERT) using the retrovirus vector pLXSN-hTERT or the lentivirus vector pCLX-PGK-hTERT vector (#114315, Addgene). Retrovirus and lentivirus preparations, infection of primary cells, and selection for immortalized cells by 100 µg/ml G418 (only with pLXSN-hTERT, 61-234, Corning) have been described previously[30]. Cells were transformed with oncogenes H-Ras[G12V], and SV40-T/t antigens expressing lentiviruses using vectors pLenti CMV-RasV12-Neo (w108-1) (HRAS[G12V], #22259, Addgene), and pLenti-CMV/TO-SV40 small + Large T (w612-1) (#22298, Addgene), respectively, as before[43]. PZP KTB40 and KTB42 cells were modified to express Tomato

Red or GFP markers using pCDH-EF1-Luc2-P2A-tdTomato (#72486, Addgene) or pCDH-EF1-Luc2-P2A-Cop GFP (#72485, Addgene) lentivirus. Cell lines in the laboratory are routinely tested for *Mycoplasma* ~every 6 months (last testing was done on February 24, 2022).

## Genetic ancestry mapping and genotype analysis

Leukocyte DNA from the same donors was obtained from the Komen Tissue Bank (KTB). Blood was collected using the BD-Vacutainer spray-coated K3EDTA tubes (Becton Dickinson). Tubes were centrifuged for 15 min at $300 \times g$. Once the upper phase (plasma) was removed, the tubes were stored in −80 °C. Peripheral blood leukocyte DNA was extracted using AutoGenprep 965 (Autogen, Inc.). Genotyping was performed using the KASP technology (LGC Genomics) and a 41-SNP panel (labeled 41-AIM panel) selected from Nievergelt et al.[84]. Genotype analysis using the 41 SNPs panel along with a Bayesian clustering method (Structure Software V2.3.4) was able to discern continental origins. We used the 940 subjects from Human Genome Diversity Project[85,86] as a reference to infer the continental origins and admixture proportions. The subjects were from 52 global populations and seven geographical regions. It is the same reference panel used by Nievergelt et al.[84]. The reference dataset includes subjects from seven continental regions, Africa, the Middle East, Europe, Central/South Asia, East Asia, the Americas, and Oceania.

## Flow cytometry analysis and cell sorting

Adherent cells were collected by trypsinization, stained using antibodies PROCR (CD201)-PE (130-105-256, Miltenyi Biotech Inc., dilution 2:200), EpCAM-APC (130-091-254, Miltenyi Biotech Inc., dilution 2:200) EpCAM-PE (130-091-253, Miltenyi Biotech Inc., dilution 2:200), PDGFRα-PE (#323506, Biolegend, dilution 5:200), CD26-APC (#563670, BD Biosciences, dilution 5:200), CD105-PE (#12-1057-42, Invitrogen, dilution 5:200), CD73-PE (561014, BD Pharmingen, dilution 5:200), CD90-APC (559869, BD Pharmingen, dilution 5:200), CD44-APC (559942, BD Pharmingen, dilution 5:200), CD24-PE (555428, BD Pharmingen, dilution 5:200), CD10-APC (#340922, BD Biosciences, dilution 5:200), CD10-PE (#340920, BD Biosciences, dilution 5:200), CD49f-APC (FAB13501A, R&D Systems, dilution 5:200), CD44-FITC (555478, BD Biosciences, dilution 5:200), CD24-FITC (#555427, BD Biosciences, dilution 5:200), CD298-FITC (130-101-291, Miltenyi Biotech Inc., dilution 5:200), ALDEFLUOR (01700; STEMCELL Technologies, dilution 5:200), and were acquired using a BD LSR II flow cytometer. Data were analyzed using FlowJo software. Forward and side scatter were used to ensure that only live cells were considered in the analysis and supplementary file contains forward and side scatter plots (Fig. S16). Gating was done using appropriate FITC (555573, BD Pharmingen, dilution 2.5:200), PE (555749, BD Pharmingen, dilution 2.5:200), and APC (555576, BD Pharmingen dilution 2.5:200) isotype control antibodies and only a representative isotype control for two fluorescent markers are shown. Samples were analyzed and sorted using BD FACSAria and SORPAria. For IL-6 signaling neutralization studies, cells were treated with anti-IL-6R antibody (MAB227, R&D Systems) or control IgG (sc-2025, Santa-Cruz Biotechnology) at 2 µg/ml concentration for 48 h prior to harvesting cells for flow cytometry.

## RNA extraction and quantitative real-time Reverse transcription polymerase chain reaction (qRT-PCR)

Total RNA was isolated using RNeasy Kit (74106, Qiagen) and 2 µg of RNA was used to synthesize cDNA with Bio-RAD iScript cDNA Synthesis Kit (170-8891, Bio-Rad). qRT-PCR was performed using Taqman universal PCR mix and predesigned gene expression assays with best coverage from Applied Biosystems. The following primer sets were used in our study: ACTB (Hs01060665_g1), ZEB1 (Hs01566408_m1),PPARG (Hs01115513_m1), RUNX1 (Hs01021971_m1), IL-6 (Hs00174131_m1), TAGLN (Hs06633192_s1), WISP1 (Hs05047584_s1), TNC (Hs01115665_m1), CSF1 (Hs00174164_m1), SPP1 (Hs00959010_m1),

IL-33 (Hs04931857_m1), DLL1 (Hs00194509_m1), F3 (Hs01076029_m1), NRG1 (Hs01101538_m1), IL-8 (Hs00174103_m1), CMTM6 (Hs00215083_m1), MIF (Hs00236988_g1), MFGE8 (Hs00983890_m1), POSTN (Hs01566750_m1).

## Generation of TMA

We created a TMA comprising healthy breast tissue (Normal-Healthy) donated to the KTB from 49 women who self-reported race and ethnicity as non-Hispanic African American and 154 women who self-reported as non-Hispanic White. The tissues were collected with written informed consent prior to donation. The age of the women who donated breast tissues to the KTB ranged from 18 to 93 years and age matched tissues were used as much as possible. The TMA also contained matched non-tumor adjacent to tumor tissue (NAT) and tumor tissue of approximately 50 donors. Part of the TMA from women of EA has been described previously[87]. Wherever possible, two cores of the same NAT and tumors were imprinted in the TMA and dot plots in the figures include H-scores and positivity data of both cores. However, for statistical analysis, data from the core with higher H-scores and positivity were used. Genetic ancestry information of Normal-Healthy donors and histopathology of tumors are shown in Fig. S4 and Table S1, respectively. Self-reported race and ethnicity was utilized from the donors of matched NATs and tumors and was not validated through genomic studies due to limited genomic DNA availability. However, for simplicity and to avoid confusion, NATs and tumors from self-reported African American and White women are labelled as AA and EA in figures and figure legends. TMA was analyzed by immunohistochemistry (IHC) for PROCR (MAB22451, R&D Systems, 10 µg/ml), ZEB1 (3G6, #14-9741-82, eBioscience, 10 µg/mL), PDGFRα (#3174, Cell Signaling Technology, 1:1000), phospho-STAT3 (#ab30647, Abcam,1:100), Estrogen Receptor alpha (ERα, Clone EP1, Dako IR 084, 15 µg/mL), GATA3 (sc-268, Santa Cruz Biotechnology,1:100), and FOXA1 (sc-6553, Santa Cruz Biotechnology, 1:100). IHC was done in a Clinical Laboratory Improvement Amendments (CLIA)-certified Indiana University Histopathology Laboratory and evaluated by three pathologists in a blinded manner. Quantitative measurements were done using the automated Aperio imaging system (ScanScope CS) and analysis was conducted using an FDA-approved algorithm. Positivity and H-scores were recorded and statistically analyzed[8]. Data were analyzed in three different ways: (i) expression differences between African ancestry and European ancestry in Normal-Healthy; (ii) expression differences between Normal-Healthy and NATs; and (iii) expression differences between NATs and tumors.

## Adipogenic and osteogenic differentiation assay

For adipogenic differentiation assay, PZP cells were plated in tissue culture dishes pre-coated with laminin-5-rich conditioned media (CM) from the 804 G cell line in low glucose DMEM supplemented with 10% FBS (#26140-079) and 10 ng/ml bFGF (#3718-FB-010) and maintained at 37 °C in a 5% CO2 incubator. Cells were allowed to grow to confluence and were then held at confluence for 2 days without changing the media prior to exposure to the adipocyte differentiation cocktail (1 µg/ml insulin, 0.25 µg/ml dexamethasone, and 0.5 mM IBMX) in fresh media without addition of bFGF. The adipocytes differentiation media was changed every 3 days until day 12 for harvest. Cells were fixed and stained with oil red O (#ECM950, Millipore) according to the manufacturer's instructions. To quantify staining, oil red O was extracted from cells using dye extraction solution (#ECM950, Millipore) according to the manufacturer's instructions and optical density was measured at a wavelength of 520 nm.

For osteogenic differentiation assay, PZP cells were plated onto tissue culture dishes in 50% low glucose DMEM/F-12 (#A4192001) supplemented with 10% FBS, 1% L-glutamine (#SCR028, Millipore Sigma). Cells were allowed to grow for 2–3 days to attain confluence, after which the normal media was removed, and osteogenic

differentiation medium was added. This media change corresponded to differentiation day 1. The osteogenic differentiation media contained 50% low glucose DMEM/F-12 supplemented with 10% FBS, 1% L-glutamine, 5 nM dexamethasone (#SCR028, Millipore Sigma), 250 μM L-ascorbic acid 2-phosphate (#SCR028, Millipore Sigma) and 10 mM β-glycerophosphate (#SCR028, Millipore Sigma). The media was changed twice a week. After 3 weeks, differentiated cells were washed in PBS and fixed in 95% methanol for 10 mins. Cells were stained with 2% alizarin red solution (Catalog no.# ECM815, Millipore) for 5 mins, then rinsed with water and imaged under fluorescence microscope to visualize the $Ca^{2+}$ deposits. To quantify $Ca^{2+}$ deposits, alizarin-red was extracted from the cells using in vitro osteogenesis assay kit (#ECM815, Millipore) according to the manufacturer's instructions and optical density was measured at a wavelength of 520 nm.

### Co-culture and cytokine/chemokine array

50% of each epithelial cell line (KTB34 and KTB39) and PZP cells (KTB40 and KTB42) were co-cultured in growth media for 2 days for RNA analysis. For cytokine/chemokine arrays, culturing of individual and co-culture was done as above and allowed to grow till the plates were confluent. Plates were washed with PBS and incubated with DMEM/F12 without other ingredients for 24 h. CM was collected and subjected to cytokine/chemokine array. The cytokine/chemokine array was performed according to the manufacturer's instructions (#ARY022B, R&D Systems).

### Antibodies and Western blotting

Primary antibodies used included rabbit anti-Ras (#3965, Cell Signaling Technologies, 1:1000), anti-pSTAT3 Y705 (#9145 S, Cell Signaling Technologies, dilution 1:1000), anti-pSTAT3 S727 (#9134D, Cell Signaling Technologies, dilution 1:1000), anti-STAT3 (#4904 S, Cell Signaling Technologies, dilution 1:1000), mouse anti-SV40-T antigens (MABF121, EMD Millipore, dilution 1:1000), and mouse anti-β-actin (A5441, Sigma-Aldrich, 1:5000). For IL-6 signaling neutralization studies, anti-IL-6R antibody (MAB227, R&D Systems) or control IgG (sc-2025, Santa Cruz Biotechnology) at 2 μg/ml concentration was added to CM for 1 h prior to adding CM to epithelial cells. Cells were treated with CM for 2 h prior to harvesting cells for western blotting. Anti-mouse (#7076, dilution 1:10000), and anti-rabbit (#7074, dilution 1:10000) horseradish peroxidase (HRP) linked secondary antibodies were purchased from Cell Signaling Technologies. Cell lysates were prepared in radioimmunoassay buffer and analyzed by western blotting[43]. Uncropped blots are presented in the Source data file.

### Xenograft study and IHC of tumors and lungs

The Indiana University Animal Care and Use Committee approved the use of animals in this study and all procedures were performed as per NIH guidelines. Transformed cells with 50% basement membrane matrix (BME) type 3 (3632-005-02, Trevigen) (total 100 μL volume) were implanted into the mammary fat pad of 5–6-week-old female NSG (NOD/SCID/IL2Rgnull) mice. A 60-days slow release 17β-estradiol (0.72 mg) pellet (SE-121, Innovative Research of America) was implanted at the time of mammary fat pad injection. Tumor growth was measured weekly and tumor volume was calculated using the formula-sagittal dimension (mm) × [cross dimension (mm)]2/2[43]. The maximal tumor burden permitted by the Indiana University Animal Care and Use Committee is 2 cm × 2 cm. The maximal tumor burden was never exceeded during this study. After 2–4 months, tumors and lungs were collected and processed for hematoxylin and eosin (H&E), ERα, GATA3, FOXA1, EpCAM (MA5-29698, Invitrogen,1:500), CK5/6 (IR 780, Dako, Ready-to-use), CK8 (35bH11, N1560, Dako, 10 μg/ml), CK14 (LL002, ab7800, Abcam, 1 μg/ml), and CK19 (IR 615, Dako, Ready-to-use) staining[43].

### Statistical analysis

Statistical analyses were conducted using Prism software program (version 9.0) and SAS software (version 9.4). Data were analyzed using T-test, one-way and two-way ANOVA. P values < 0.05 were considered statistically significant. In case of TMA results, nonparametric Wilcoxon rank-sum tests were used for unpaired analysis, as positivity and H-scores were not normally distributed[8].

### Reporting summary

Further information on research design is available in the Nature Portfolio Reporting Summary linked to this article.

## Data availability

This paper does not report any high throughput data. All reagents created as part of the paper will be distributed upon meeting requirements of the Indiana University. Indiana University requires recipient scientists and institutions to sign a material transfer agreement. Requests about access to data or for reagents, including cell lines and TMA, should be submitted to the corresponding author (hnakshat@iupui.edu). Source data are provided with this paper. Source data file is also available through Figshare https://doi.org/10.6084/m9.figshare.24061173.

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

## Acknowledgements

We thank members of the Indiana University Simon Comprehensive Cancer Center (IUSCCC) flow cytometry core, animal facility, and tissue procurement cores and the Susan G Komen Tissue Bank for various tissues and reagents. We also thank members of the Komen Tissue Bank and IUSCCC tissue bank including Jill Henry, Mary Cox, Rana German, Rockey Pam, and Julia von Arx for their work in generating TMA and associated data. We thank Dr. Natascia Marino for contributing part of TMA containing breast tissues of women of European ancestry. Special thanks to Dr. Christine Ambrosone for advice related to genetic ancestry mapping and immune cells. We thank countless number of women for donating their breast tissue for research purpose as well as volunteers who facilitated tissue collection. This work is supported by DOD-W81XWH-15-1-0707, DOD-W81XWH-20-1-0577, Susan G. Komen for the Cure (DRS20645418) and 100 Voices of Hope to HN. Susan G. Komen for the Cure, Breast Cancer Research Foundation and Vera Bradley Foundation for Breast Cancer Research provide funding support to Komen Normal Tissue Bank.

## Author contributions

B.K., and H.N. conceived and designed research. B.K., A.S.K., P.B.N., L.A.B., C.J.T., A.M.S., and H.N. performed experiments. B.K., A.S.K., K.B., P.B.N., M.M.G., R.J.A., M.S., S.K.A., K.D.M., and H.N. acquired data. B.K., J.G., and M.L.C did genetic ancestry analysis. B.K., G.S., and H.N. interpreted data. B.K., A.S.K., and H.N. wrote the paper. H.N., A.M.S., P.B.N. Administrative, technical, or material support. H.N. Study supervision. All authors reviewed the paper and contributed valuable inputs.

## Competing interests

The authors declare no competing interests.
