## [Peer Review File · Nature Communications]

REVIEWER COMMENTS

Reviewer #2 (Remarks to the Author): Expert in breast cancer development, stem cells, and transdifferentiation

This is an intriguing study demonstrating the prevalence of a ZEB1+ stromal cell population in the normal breast tissue of women of African Ancestry (AA) compared to European Ancestry (EA). The authors showed that these cells also express PROCR and PDGFR α , thus referring to them as PZP cells. They further demonstrated that PZP cells resemble mesenchymal stem cells and could trans-differentiate into adipocytes or osteocytes. Upon co-culture with epithelial cells, the co-culture led to increased production of IL-6 and triggered luminal-to-basal epithelial cell fate change. Upon transformation by HRasG12V \pm SV40-T/t antigens, PZP cells could generate metaplastic carcinoma in NSG mice. The authors hypothesized that this stromal cell population is a result of the genetic ancestry-dependent variability (i.e., AA vs. EA) and constitutes a biological basis for the higher mortality from TNBC in women of AA.

However, the manuscript is largely descriptive in nature and could be improved by addressing the following questions:

1. When characterized PROCR expression in TMA, its expression was observed in both ductal and stromal cells (Fig. 4). The authors reported that PROCR-expressing cells were enriched in the KTB-normal breast tissue of AA women compared with EA and LA women (Fig. 4B, Table 1). It is unclear if the authors subdivide these PROCR-expressing cells to ductal cells and stromal cells, which subset of the PROCR-expressing cells in the KTB-normal breast tissue of AA women would be more abundant than those from EA and LA women.
2. The IHC data in Fig. 6 is a bit confusing. The quantification data in Fig. 6C showed a higher level of PDGFR α positivity in KTB-normal breast tissue of AA women compared with EA and LA women, but in the representative IHC staining pictures shown in Fig. 6A, the signal for PDGFR α appeared weaker in the AA sample than those in the EA and LA sample.
3. The identification of increased expression/production of IL-6 upon co-culture of PZP cells and epithelial cells is intriguing. However, since this was only observed in the co-culture of one PZP cell line and one immortalized luminal epithelial cell line (n=1), the finding should be further validated (in addition to phosphor-STAT3 staining). At the functional level, the authors implied that this communication (i.e., IL-6 production) might result in luminal epithelial cells acquiring basal/stem cell characteristics. It would be very helpful if this could be tested functionally (e.g., by using IL-6 neutralizing antibody or inhibitor to block its activity in the co-culture, one can test if this treatment inhibits acquisition of basal phenotype by luminal cells).

4. The authors implied a potential role of PZP cells in immune cell modulation in the microenvironment based on analysis of various candidate immunomodulatory genes or cytokine genes in PZP cells (e.g., under the setting of PZP-epithelial co-culture). It would be helpful if the authors could perform some initial validations in their TMA (e.g., AA vs. EA and LA normal breast tissues) for various immune cell populations and/or their functional state.

5. PZP cell lines are immortalized primary cells and therefore even within each cell line, the cell line may include both stromal cells and a small number of epithelial cells. Therefore, in both the co-culture and HRasG12V transformation experiments, even though the conclusion is that PZP (stromal) cells could potentially acquire epithelial characteristics, one cannot rule out the possibility that the fraction of cells that have acquired epithelial phenotype is due to expansion of a rare subpopulation of epithelial cells pre-existing in the PZP cell lines. The authors emphasized KTB42, which exhibited a more profound phenotype of acquisition of epithelial phenotype. In fact, in Fig. 3H, it was shown that KTB42 has 1.92% of CD44+CD24+ cells; in Fig. 1B, KTB42 has 1.93% of EpCAM+ cells, raising a possibility that there are pre-existing epithelial cells in the KTB42 cell line and these CD44+CD24+ or EpCAM+ cells got expanded, which gave rise to the “acquired” epithelial phenotype.

6. In the “Materials and Methods” section, the authors mentioned that “tissues from AA donors yielded both loosely attached elongated cell clusters and cuboidal epithelial colonies. Differential trypsinization was used to separate two population of cells for immortalization”. It is unclear whether the PZP cell lines were generated from the loosely attached elongated cell clusters or the cuboidal epithelial colonies, or both.

Minor points:

1. Page 13, “Morphologically, PROCR+/EpCAM- cell lines showed features of epithelial to mesenchymal transition (EMT), which is expected based on ZEB1 expression (Fig. S2B)”. The mentioning of EMT here is premature, as if these PROCR+/EpCAM- cells are stromal cells in nature, then they are mesenchymal cells and EMT doesn’t apply to them; if they are derived from epithelial cells, then EMT applied to them, but the origin of these PROCR+/EpCAM- cells is not clear.

2. Page 30, “Our results indicate that PZP cells are the cell-of-origin of this cancer type”. The xenograft experiment in Fig. 9 suggested that PZP cells could give rise to metaplastic carcinoma, but this doesn’t rule out that metaplastic carcinoma could also come from transformation of luminal or basal epithelial cells.

Reviewer #3 (Remarks to the Author): Expert in breast cancer genomics, functional genomics, ancestries, bioinformatics, and tumour microenvironment

Kumar and collaborators extensively characterized PZP cell lines from healthy women of AA ancestry and compared them with other ancestries, as a result they found several ancestry-dependent characteristics. In summary, their findings suggest that these cells could resemble mesenchymal stem/stromal cells enriched for transdifferentiation capabilities and tumorigenic features. In vitro experiments and immunochemical characterization of human samples have also shown that PZP cells can regulate signaling in other stromal cells. Undoubtedly, having a novel representation of a minority population such as African women is relevant and, in general, a great effort of the authors is demonstrated through the results, but, I think there are some important points that need to be addressed.

Major and minor comments

1. In general, following the full story through the text is difficult, so I'll suggest clearly stating the conclusions, why the strategy used was determined, in short presenting a smoother story that would be easier to read.
2. Figure one, change the color legend of Oceania as it is almost the same as Africa
3. Improve quality of figure 2 and in general of all the flow cytometry scatterplots
4. Why are the results shown in Figure 2 only performed on the KTB40, KTB42 and KT32? I think it is important to perform the characterization on the complete samples available to obtain a more robust characterization.
5. What is the transdifferentiation efficiency into adipogenic and osteogenic lineages of immortalized PROCR±/EpCAM+ cell variants? If the authors' hypothesis is correct, PROCR+/EpCAM- cells would have higher transdifferentiation properties.
6. Is there any way to quantify the mineralization of the matrix with Ca²⁺, thus accompanying the images presented with the results of the quantification.
7. Figure 3 shows a heterogeneous population: while 50% of the evaluated cells presented the described phenotype, the other 50% did not, please describe and discuss this scenario.
8. Indicate with a dotted line in figures 3A and 3B the means
9. Standardize the nomenclature of the figure Fig. 3D – G referring to CD73+ as in the text or CD73^{high} as in the figure.
10. In figure 3d-g although it is true that there is variability in the population phenotype with respect to the stem cell marker CD90/CD73, a trend towards CD90^{high}/CD73+ enrichment is also evident, please discuss this.
11. FIGURE 3H-I Tumor-initiating cells: the combination of a high ratio of CD44/CD24 and ALDH1+ would be a more reliable way to characterize CSCs

12. Thinking about this author's conclusion: "However, the lack of EpCAM expression in PZP cells suggests that not all CD44+/CD24- cells in the breast, including breast tumors, have stem cell properties." I would like to comment that CD44+/CD24- cells have been identified with luminal epithelial cells that resemble progenitor cells. On the other hand, myoepithelial cells and basal progenitor cells are defined by an EpCAM-/CD49f+ phenotype. Therefore, stem cell-like properties are also detected in this cellular compartment and do not necessarily exclude its stem cell properties.13. In Figure 4: A-B) Place the zoom detail close to the zoom out image so it is clear which field is being magnified for a clearer view

15. Based on this general idea presented by the authors "Thus, ancestry-dependent variability in stromal PZP cell levels has an impact on normal and cancerous breast biology and this suggests that mammary epithelial cell biology between these different groups, which may contribute to altered susceptibility to tumor initiation, progression, and/or metastasis." How do you reconcile this author's conclusion with the following fact:

it is true that normal healthy tissue from AA over-expressed PROC in comparison to EA, but when looking at the NAT tissue there is not a significant differences in PROC levels between both populations, So what might this increased level of PROC in healthy tissue implicate in cancer development?. Similarly, behavior is identified in ZEB1 evaluation. Although, genetic ancestry-dependent differences were identified I cannot get their influences in tumor cells and tumor adjacent tissue by ancestry variation.

16. For a clearer nomenclature, I suggest renaming normal tissue from healthy volunteers as Normal-healthy, NAT as non-tumor adjacent to tumor tissue, primarily in light of the authors' conclusion that "NATs are not normal" based on some protein expression pattern

17. Certainly Figure 7 is difficult to interpret since the two models of epithelial cell lines clearly show an opposite behavior in some of the genes evaluated as TNC and Il-33 which show an individual behavior. This heterogeneity was also exhibited by the cocultured cells, for example, KTB40 seems to generate a much more pronounced effect.

18. Metaplastic breast carcinoma, although rare, is more frequent in AA women than in EA women, and authors reported that PZP cells are the cell of origin of this type of cancer. Due to its incidence, I suppose that this subtype is poorly represented in the authors' TMA, so the phenotype described could be extended or present not only to this hispoligal subtype, nut also, to some others as basals. 19. Please discuss this and include the clinopathological table of the samples evaluated by TMA.

20. How the stem-like phenotype is represented in the TMA samples??

21. In addition to the biological relevance of RAS in tumorigenesis, mainly in certain subtypes of breast cancer, what are the reasons for inducing this signaling pathway in particular? Is there an enrichment in RAS activation in PZP cells?

22. As already mention one of my main constrains is that the results, as expected ny the very limited number of avaluted samples, are really heterogenous. The authors sometimes described the anecdotic result of one PZP cell sample and generalice the observed phenotype. For example: "Transformation of PZP cells with activated HRasG12V increased the fraction of cells that have acquired epithelial

phenotype and express EpCAM, particularly in KTB42 “, this statement is only true for one sample out of 3, in the other 2 samples this behavior is in fact over the overexpression of mutant HRasG12V . many of the results are related only with one sample, not Even the majority of the samples,

As I already mentioned, one of my main objections is that the results, as expected due to the very limited number of samples evaluated, are really heterogeneous. The authors sometimes described the anecdotal result of one PZP cell sample and generalized the observed phenotype to all the phenomena. For example: “Transformation of PZP cells with activated HRasG12V increased the fraction of cells that have acquired epithelial phenotype and express EpCAM, particularly in KTB42”, this statement is only true for one sample out of 3, in the other 2 samples this behavior is in fact, opposite. Many of the results are related to only one sample, nut presented as the general phenotype.

23. Describe results regarding SV40-T/t antigen and their combination

24. Include y axis tittle in Fig Sup 1

25. Indicate the total numbers of evaluates samples in each plot

26. Be more moderate in generalizing this phenomenon to women with AA ancestry, since their strategy does not allow their conclusions to be extended to the population level

Reviewer #4 (Remarks to the Author): Expert in breast cancer genetics and epidemiology, ancestries, and health disparities

My comments are focused on terminology around race/ethnicity and ancestry as well as on genetic ancestry estimation and the use of these estimates in the analyses conducted.

As a general comment I would like to point out that the use of the term Latina or Latino ancestry is confusing. Ancestry (commonly implying genetic ancestry) usually refers to the continental origin of ancestors. A large proportion of individuals who self-identify as Hispanic/Latino are of European Ancestry, Indigenous American, African or Asian ancestry (to a lesser extent), or of mixed European, Indigenous American, African, and Asian ancestry. If the authors are using self-identity based on census categories, for example African American/Black, non-Hispanic White, Hispanic/Latino, then they should explicitly say so. If they are referring to genetic ancestry, then they can say African, European, Indigenous American, East or South Asian, etc. But not Latina Ancestry. Related to this, Table 1 includes

'race' in the title but then proceeds to include African American, European, Latino. These are not conventional categories and authors should try to be precise. If they want to use the census categories then it should be racial and ethnic identity and should include "African American/Black", "non-Hispanic White" and "Hispanic/Latino".

Both intro and discussion sections are lacking proper contextualization of the study within the large amount of research that has been conducted investigating the higher risk of breast cancer mortality among African American/Black women. There are studies looking at social determinants as well as biological determinants linked to ancestry. As rationale for exploring the association between African ancestry and molecular biology of normal and cancer cells the authors cite a paper that reported genetic differences between people of African ancestry and other groups, as if this were recent information for the scientific community, which is not. The foundation for this paper should better acknowledge the work done by others demonstrating that disparities affecting African American/Black women have not been fully explained and that prior research has shown biological differences when comparing biology of bio specimens derived from individuals from different race/ethnicity category.

Regarding genetic ancestry estimates:

The panel of 41 SNP does not 'discern' the continental origins of participants, it provides an estimation of ancestry proportions. Using a small panel such as the one described, credible intervals for the estimates could be informative. Further specification of parameters set for the process of estimating ancestry proportions is needed. For ancestry estimation with the small number of 41 aims authors used 1000G data. However, ideally, they should have used a large non-admixed reference set as the authors of the original 41-aim panel paper described. Potential issues in the ancestry estimations in the current article are reflected by the high Central South Asia estimates for the Hispanic/Latina samples. The average Central South Asia estimate for Hispanic/Latinos is usually very low (ranging from 1 to 4%), and lower than the Indigenous American component of ancestry (in their paper referred to 'American'). However, in this article, the proportion of estimated Central South Asian ancestry is higher than that of Indigenous American ancestry. Authors should have provided mean and SD measures for ancestry estimates instead of just figures.

The samples used in the study seem to have variation in ancestry proportions. Hence, the authors could have conducted studies using samples with similar proportion of African and European ancestry, independently of race/ethnic category of the participant and contrast results using the categories. This could give important information about ancestry vs. race/ethnicity category effects in the results they report. What is the correlation between the biological markers measured in the paper and African ancestry proportion? For example, is there a correlation between level of enrichment of PROCR+/EpCAM- cells and proportion of African Ancestry? KTB40 and KTB32 have quite a lot of non-African ancestry influence based on Figure 1 A. KTB59 is mostly African ancestry. Are levels between KTB40 and KTB32 more similar and very different from KTB59? Sample size is relatively small overall, but for some of these analyses authors could look at ancestry proportion to provide more support for a

genetic-ancestry biological difference as opposed to a biological difference driven by biological effects related to race/ethnicity category experiences.

Response to Review:

Reviewer #2 (Remarks to the Author): Expert in breast cancer development, stem cells, and transdifferentiation

This is an intriguing study demonstrating the prevalence of a ZEB1+ stromal cell population in the normal breast tissue of women of African Ancestry (AA) compared to European Ancestry (EA). The authors showed that these cells also express PROCR and PDGFR α , thus referring to them as PZP cells. They further demonstrated that PZP cells resemble mesenchymal stem cells and could trans-differentiate into adipocytes or osteocytes. Upon co-culture with epithelial cells, the co-culture led to increased production of IL-6 and triggered luminal-to-basal epithelial cell fate change. Upon transformation by HRasG12V \pm SV40-T/t antigens, PZP cells could generate metaplastic carcinoma in NSG mice. The authors hypothesized that this stromal cell population is a result of the genetic ancestry-dependent variability (i.e., AA vs. EA) and constitutes a biological basis for the higher mortality from TNBC in women of AA.

However, the manuscript is largely descriptive in nature and could be improved by addressing the following questions:

1. When characterized PROCR expression in TMA, its expression was observed in both ductal and stromal cells (Fig. 4). The authors reported that PROCR-expressing cells were enriched in the KTB-normal breast tissue of AA women compared with EA and LA women (Fig. 4B, Table 1). It is unclear if the authors subdivide these PROCR-expressing cells to ductal cells and stromal cells, which subset of the PROCR-expressing cells in the KTB-normal breast tissue of AA women would be more abundant than those from EA and LA women.

Response: We recalculated PROCR and PDGFR α in epithelium and stroma of normal breast tissues separately and data are shown in Figure S5. Stromal cells expressed ~5-folds higher levels of PROCR compared to epithelium (H-score values 84.52 versus 15.48). Stromal levels of PROCR was higher in African American compared to European ancestry and IA samples, although differences did not reach statistical significance ($p=0.0894$). One reason may be due to reduced number of samples among African Americans with both stromal and epithelial expression of PROCR (N=32). With respect to PDGFR α , stroma expressed ~40-folds higher levels compared to epithelial cells. Epithelial cell expression of PDGFR α was significantly higher in African American samples compared to European ancestry samples ($p=0.0001$, 14.09 vs 1.09) but not in IA samples. However, these results need to be interpreted with caution because number of samples with both epithelial and stromal expression among African Americans reduced to 19. Therefore, we have not elaborated these results much further in the manuscript.

2. The IHC data in Fig. 6 is a bit confusing. The quantification data in Fig. 6C showed a higher level of PDGFR α positivity in KTB-normal breast tissue of AA women compared with EA and LA women, but in the representative IHC staining pictures shown in Fig. 6A, the signal for PDGFR α appeared weaker in the AA sample than those in the EA and LA sample.

Response: Apologies for not providing truly representative IHC image. We have retaken pictures of IHC images and replaced Figure 6A with a better representative image.

3. The identification of increased expression/production of IL-6 upon co-culture of PZP cells and epithelial cells is intriguing. However, since this was only observed in the co-culture of one PZP cell line and one immortalized luminal epithelial cell line (n=1), the finding should be further validated (in addition to phospho-STAT3 staining). At the functional level, the authors implied that this communication (i.e., IL-6 production) might result in luminal epithelial cells acquiring basal/stem cell characteristics. It would be very helpful if this could be tested functionally (e.g., by using IL-6 neutralizing antibody or inhibitor to block its activity in the co-culture, one can test if this treatment inhibits acquisition of basal phenotype by luminal cells).

Response: Results of IL-6R neutralizing antibody experiments on cell surface marker profiles and phospho-STAT3 levels are shown in Figure S11-S12. We examined the effects of IL-6R antibodies on CD49f/EpCAM (Figure S11), CD44/EpCAM (Figure S11), CD10/CD24 (Figure S12), and phospho-STAT3 status (Figure 8). The most significant effects of IL-6R neutralizing antibodies were observed on phospho-STAT3 levels. As shown in Figure 8F, conditioned media from the luminal epithelial cell lines KTB34, PZP cell line KTB40 and from co-cultured KTB34+KTB40 cell lines increased pSTAT3 in KTB34. STAT3 phosphorylation was significantly higher when media from KTB34+KTB40 co-cultured cells were used. STAT3 phosphorylation by KTB40 and KTB34+KTB40 cultured cells was inhibited by IL-6R antibody. As demonstrated in the original submission, co-culturing of epithelial cell lines (KTB34, a luminal-like cell lines and KTB39, a normal-like cell line, Kumar et al., Cancer Research, 78:5107-5123) with PZP cells increased the number of CD49f+/EpCAM- cells (Figure 8 and Figure S11B). IL-6R antibody reduced the levels of CD49+/EpCAM- cells under co-culture conditions, particularly with luminal like KTB34 cells with a statistical significance ($p=0.0121$). PZP cells also increased the number of CD44+/EpCAM- cells (Figure S11) and reduced the levels of CD24+/CD10^{high} KTB34 and KTB39 cells (Figure S12B). However, these effects of PZP cells were unaffected by the anti-IL-6R antibody.

Liu et al have previously demonstrated that ALDEFLUOR+ cells in normal breast correspond to epithelial-like stem cells, whereas CD44+ cells are considered basal-like stem cells of the breast (Stem Cell Reports 2:78-91). We observed PZP cells reducing the number of ALDEFLUOR+ subpopulation of KTB34 and KTB39 cells (Figure S10). These results coupled with increase in CD49+/EpCAM- and CD44+/EpCAM- cells upon co-culturing of epithelial cells with PZP cells clearly demonstrate the effects of PZP cells on luminal cells acquiring basal-like characteristics. We suggest that these effects of PZP cells is partially dependent on IL-6.

4. The authors implied a potential role of PZP cells in immune cell modulation in the microenvironment based on analysis of various candidate immunomodulatory genes or cytokine genes in PZP cells (e.g., under the setting of PZP-epithelial co-culture). It would be helpful if the authors could perform some initial validations in their TMA (e.g., AA vs. EA and LA normal breast tissues) for various immune cell populations and/or their functional state.

Response: We agree with the reviewer that such an analysis is highly meaningful. We had previously performed immune cell analyses by IHC of TMAs containing normal, tumor adjacent-normal and tumors of women of African and European ancestry (Clinical Cancer Research 25:2848-2859). However, measurable levels of immune cells were found only in

tumor-adjacent and tumors but not in normal breast tissues. Tumor adjacent normal tissues of women of African ancestry contained elevated levels of CD8+ T cells and PD1+ cells compared to tumor adjacent normal tissues of women of European ancestry. Tumor adjacent normal tissues of women of European ancestry contained elevated number of PD-L1+ cells. No differences were noted with CD68+ macrophages. Only time we could detect immune cells in the normal breast tissues was with single cell RNA sequencing (Cell Reports Medicine 2:100219), which requires large amount of tissues. Thus, because of lower sensitivity of IHC-based assays to measure immune cells in the normal breast tissues, we are unable to generate these data.

5. PZP cell lines are immortalized primary cells and therefore even within each cell line, the cell line may include both stromal cells and a small number of epithelial cells. Therefore, in both the co-culture and HRasG12V transformation experiments, even though the conclusion is that PZP (stromal) cells could potentially acquire epithelial characteristics, one cannot rule out the possibility that the fraction of cells that have acquired epithelial phenotype is due to expansion of a rare subpopulation of epithelial cells pre-existing in the PZP cell lines. The authors emphasized KTB42, which exhibited a more profound phenotype of acquisition of epithelial phenotype. In fact, in Fig. 3H, it was shown that KTB42 has 1.92% of CD44+CD24+ cells; in Fig. 1B, KTB42 has 1.93% of EpCAM+ cells, raising a possibility that there are pre-existing epithelial cells in the KTB42 cell line and these CD44+CD24+ or EpCAM+ cells got expanded, which gave rise to the “acquired” epithelial phenotype.

Response: We have addressed these concerns by performing two sets of experiments and results are presented in Figure S15. We used KTB42 cell line for this purpose as this PZP cell line showed highest level of plasticity. In first series of experiments, we generated KTB42 RAS+SV40 transformed cells and sorted PROCR+/EpCAM- cells by flow cytometry, Please see the gate used to sort these cells (Figure S15A). We made sure that gate is far away from any possible contamination of EpCAM+ cells. After expansion and ensuring that cells remain EpCAM-, we transplanted cells into NSG mice and characterized tumors by histology. The resulting tumors were spindle shaped anaplastic carcinomas with pockets of EpCAM+ tumor cells and the entire tumor stained for PROCR (Figure S15B). In the second series of experiments, single cells of PROCR+/EpCAM- Ras+SV40 transformed KTB42 cells were plated into 96 well plate and a single cell-derived clone was generated. Cells from the single cell-derived clone were mostly PROCR+/EpCAM-. Tumor-derived from this single cell clone also generated spindle cell containing carcinomas with pockets of EpCAM+ tumor cells. From these results, we suggest that transformed PZP cells can transdifferentiate into epithelial cells and our results are unlikely due to few EpCAM+ cells contaminating PZP cell lines.

6. In the “Materials and Methods” section, the authors mentioned that “tissues from AA donors yielded both loosely attached elongated cell clusters and cuboidal epithelial colonies. Differential trypsinization was used to separate two population of cells for immortalization”. It is unclear whether the PZP cell lines were generated from the loosely attached elongated cell clusters or the cuboidal epithelial colonies, or both.

Response: When we plated cells generated from breast tissues of African American donors but not European donors, we always observed two types of colonies- one elongated cell clusters without cell-cell contact and the other cuboidal epithelial cell clusters with robust cell-cell

contact within first five days of plating. Elongated cell clusters were seen with tissues from European donors only after two weeks culturing mostly in cases without any epithelial cell colonies. We separate PZP cells from epithelial cells by trypsinizing cells at room temperature for two minutes and collecting supernatant with detached cells. Adherent epithelial cell colonies with cell-cell contact needs at least 15 minutes of trypsinization to come off plates. Thus, PZP cell are derived from loosely attached elongated cell colonies.

Minor points:

1. Page 13, “Morphologically, PROCR+/EpCAM- cell lines showed features of epithelial to mesenchymal transition (EMT), which is expected based on ZEB1 expression (Fig. S2B)”. The mentioning of EMT here is premature, as if these PROCR+/EpCAM- cells are stromal cells in nature, then they are mesenchymal cells and EMT doesn’t apply to them; if they are derived from epithelial cells, then EMT applied to them, but the origin of these PROCR+/EpCAM- cells is not clear.

Response: We agree with the comments from the reviewer and removed reference to EMT. Since PZP cells are also CD73+, a specific marker of mesenchymal stem cells, referring these cells are mesenchymal cells is appropriate.

2. Page 30, “Our results indicate that PZP cells are the cell-of-origin of this cancer type”. The xenograft experiment in Fig. 9 suggested that PZP cells could give rise to metaplastic carcinoma, but this doesn’t rule out that metaplastic carcinoma could also come from transformation of luminal or basal epithelial cells.

Response: We agree with reviewer’s word of caution with respect to our interpretation. We have modified the sentence to indicate that PZP cells could be one of cells-of-origin of metaplastic carcinomas.

Reviewer #3 (Remarks to the Author): Expert in breast cancer genomics, functional genomics, ancestries, bioinformatics, and tumour microenvironment

Kumar and collaborators extensively characterized PZP cell lines from healthy women of AA ancestry and compared them with other ancestries, as a result they found several ancestry-dependent characteristics. In summary, their findings suggest that these cells could resemble mesenchymal stem/stromal cells enriched for transdifferentiation capabilities and tumorigenic features. In vitro experiments and immunochemical characterization of human samples have also shown that PZP cells can regulate signaling in other stromal cells. Undoubtedly, having a novel representation of a minority population such as African women is relevant and, in general, a great effort of the authors is demonstrated through the results, but, I think there are some important points that need to be addressed.

Mayor and minor comments

1. In general, following the full story through the text is difficult, so I'll suggest clearly stating the conclusions, why the strategy used was determined, in short presenting a smoother story that would be easier to read.

Response: Thank you for your suggestion. We have modified the manuscript and we hope that now the conclusions are easy to follow. Title of the manuscript has been modified and a schematic view of the manuscript is included as Figure 10H.

2. Figure one, change the color legend of Oceania as it is almost the same as Africa

Response: We have changed the color for Oceania in the revised manuscript.

3. Improve quality of figure 2 and in general of all the flow cytometry scatterplots

Response: The original submission contained low resolution figures because the manuscript exceeded 40 MB. The revised version has high resolution scatterplots, which show clear patterns.

4. Why are the results shown in Figure 2 only performed on the KTB40, KTB42 and KT32? I think it is important to perform the characterization on the complete samples available to obtain a more robust characterization.

Response: Figures 2, 3, and Figure S1 now contain data from six additional PZP cell lines (total 9 cell lines from 9 different donors). All cell lines show similar cell cycle markers profiles, with the exception of CD73/CD90 staining patterns, which showed inter-individual heterogeneity. Six cell lines examined showed adipogenic and osteogenic differentiation capabilities. Therefore, we now believe that data are robust.

5. What is the transdifferentiation efficiency into adipogenic and osteogenic lineages of immortalized PROCR±/EpCAM+ cell variants? If the authors' hypothesis is correct, PROCR+/EpCAM- cells would have higher transdifferentiation properties.

Response: We have examined osteogenic and adipogenic differentiation capabilities of PROCR±/EpCAM+ KTB34 and KTB39 cell lines (Figure 2). Although these cell lines showed low levels of osteogenic differentiation, extent of differentiation was significantly lower than that of PZP cell lines (Figure 2B and F).

6. Is there any way to quantify the mineralization of the matrix with Ca²⁺, thus accompanying the images presented with the results of the quantification.

Response: Quantitation of Oil Red O for adipogenic differentiation and Alizarin Red staining for osteogenic differentiation is shown in Figure 2B and Figure 2F, respectively.

7. Figure 3 shows a heterogeneous population: while 50% of the evaluated cells presented the described phenotype, the other 50% did not, please describe and discuss this scenario.

Response: Based on CD73, CD90, CD105, and CD26 staining patterns, we suspect that PZP cells have some similarity to mesenchymal stem cells with significant transdifferentiation capabilities towards lobular and interlobular fibroblast lineages (based on CD105 and CD26 staining pattern), osteogenic and adipogenic lineages. We attempted to sort PZP cells into three groups based on CD73/CD90 staining pattern (boxes in Figure 3D) and sorted cells did not proliferate well for us to perform any functional studies. Heterogeneity of PZP cells is discussed in pages 9, 10 of the revised manuscript.

8. Indicate with a dotted line in figures 3A and 3B the means.

Response: These figures have been modified to include suggested information.

9. Standardize the nomenclature of the figure Fig. 3D – G referring to CD73+ as in the text or CD73high as in the figure.

Response: We apologize for these errors and we have changed nomenclature to CD73high in the revised manuscript.

10. In figure 3d-g although it is true that there is variability in the population phenotype with respect to the stem cell marker CD90/CD73, a trend towards CD90high/CD73+ enrichment is also evident, please discuss this.

Response: We have discussed potential mesenchymal stem cell like properties of PZP cells based on CD73 and CD90 staining pattern in the revised manuscript page 10.

11. FIGURE 3H-I Tumor-initiating cells: the combination of a high ratio of CD44/CD24 and ALDH1+ would be a more reliable way to characterize CSCs

Response: We have performed ALDEFLUOR assay with PZP cells and results are shown in Figure S3. Less than 5% of cells were ALDEFLUOR+. This is in contrast to PROCRA±/EPCAM+ KTB34 and KTB39 cells, which contained ~30% ALDEFLUOR+ cells.

12. Thinking about this author's conclusion: "However, the lack of EpCAM expression in PZP cells suggests that not all CD44+/CD24- cells in the breast, including breast tumors, have stem cell properties." I would like to comment that CD44+/CD24- cells have been identified with luminal epithelial cells that resemble progenitor cells. On the other hand, myoepithelial cells and basal progenitor cells are defined by an EpCAM-/CD49f+ phenotype. Therefore, stem cell-like properties are also detected in this cellular compartment and do not necessarily exclude its stem cell properties.

Response: We agree with reviewers comments and made necessary changes to the text taking into consideration Stem Cell Report publication by Dr. Mike Wicha's group (2:78-91). However, we note that PZP cells, despite being CD44+/CD24-, are CD49f- and thus are not similar to basal progenitor cells. These cells acquired CD49 expression only upon transformation with Ras oncogene (Fig. 10).

13. In Figure 4: A-B) Place the zoom detail close to the zoom out image so it is clear which field is being magnified for a clearer view

Response: Thank you for this suggestion. We have provided necessary details in figures included in the revised manuscript.

15. Based on this general idea presented by the authors “Thus, ancestry-dependent variability in stromal PZP cell levels has an impact on normal and cancerous breast biology and this suggests that mammary epithelial cell biology between these different groups, which may contribute to altered susceptibility to tumor initiation, progression, and/or metastasis.” How do you reconcile this author's conclusion with the following fact:

it is true that normal healthy tissue from AA over-expressed PROC in comparison to EA, but when looking at the NAT tissue there is not a significant differences in PROC levels between both populations, So what might this increased level of PROC in healthy tissue implicate in cancer development?. Similarly, behavior is identified in ZEB1 evaluation. Although, genetic ancestry-dependent differences were identified I cannot get their influences in tumor cells and tumor adjacent tissue by ancestry variation.

Response: Our suggestion is based on differences in basal levels of PZP cells in AA compared to EA. Because of higher basal levels of PZP cells, tumor initiating events in AA can already take advantage of PZP cells to promote tumor growth. As shown in Figures 7 and 8, PZP cell-epithelial cell interaction leads to elevated levels of IL-6 and STAT3 activation in epithelial cells, which AA has an advantage over EA. In case of EA, tumor progression will have a lag period requiring tumor cell and residual PZP cell interaction to promote proliferation of PZP cells. We have provided this explanation in the revised manuscript (page 25).

16. For a clearer nomenclature, I suggest renaming normal tissue from healthy volunteers as Normal-healthy, NAT as non-tumor adjacent to tumor tissue, primarily in light of the authors' conclusion that "NATs are not normal" based on some protein expression pattern

Response: Thank you for this suggestion, which has been incorporated in the revised manuscript.

17. Certainly Figure 7 is difficult to interpret since the two models of epithelial cell lines clearly show an opposite behavior in some of the genes evaluated as TNC and Il-33 which show an individual behavior. This heterogeneity was also exhibited by the cocultured cells, for example, KTB40 seems to generate a much more pronounced effect.

Response: KTB40 and KTB42 are derived from different donors and as shown in Figure 3, differ with respect to CD73/CD90 staining pattern. In our experience, KTB40 grows faster compared to KTB42, which may have contributed to differences in growth under co-culture conditions and results shown in Figure 7. We have included this explanation in the revised manuscript. It is extremely difficult to create experimental conditions to have two cell lines grow at a similar rate.

18. Metaplastic breast carcinoma, although rare, is more frequent in AA women than in EA women, and authors reported that PZP cells are the cell of origin of this type of cancer. Due to its incidence, I suppose that this subtype is poorly represented in the authors' TMA, so the

phenotype described could be extended or present not only to this hispoligal subtype, nut also, to some others as basals.

Response: As noted by the reviewer, metaplastic cancers are rare and we do not have any of this tumor type in our TMA. Based on new data presented in Figure S15, PZP cells could generate mixed carcinoma subtypes with some basal cell features. Because this is simply a speculation, we did not elaborate it further in the manuscript.

19. Please discuss this and include the clinopathological table of the samples evaluated by TMA.

Response: Table S1 describes clinicopathological features of TMA.

20. How the stem-like phenotype is represented in the TMA samples??

Response: This is a difficult question to answer. However, since ZEB1 is expressed only in stromal cells, we can conclude that ZEB1+ cells are PZP cells. We do not want to provide wrong assessment of stem cells within tumors through our analysis. Unfortunately, because most samples in our TMA are from the normal breast with limited tissues, it is difficult to stain for additional markers with too many sample drop off.

21. In addition to the biological relevance of RAS in tumorigenesis, mainly in certain subtypes of breast cancer, what are the reasons for inducing this signaling pathway in particular? Is there an enrichment in RAS activation in PZP cells?

Response: We have attempted several combinations of oncogenes in our in vitro transformation assays but only combinations with RAS oncogene provided consistent transformation. Although long considered irrelevant, it has now become apparent that RAS signaling aberration accounts for ~12% of metastatic breast cancers. Using cBioportal, we have uncovered that KRAS, HRAS, NF-1 (negative regulator of RAS) and RASA1 (negative regulator of RAS) mutations in 12% metastatic breast cancers. To determine whether RAS signaling pathway is more active in PZP cells compared to epithelial cells, we used our RNA-seq data of KTB40 and KTB42 cells and compared them with RNA-seq data of six epithelial cell lines. Differentially expressed genes were subjected to Ingenuity pathway analysis. We did not observe RAS pathway activation in PZP cells compared to epithelial cells. However, we did observe activation of osteogenic pathway in PZP cells compared to epithelial cells, which is consistent with their ability to undergo osteogenic differentiation.

22. As already mention one of my main constrains is that the results, as expected ny the very limited number of avaluted samples, are really heterogenous. The authors sometimes described the anecdotic result of one PZP cell sample and generalice the observed phenotype. For example: “Transformation of PZP cells with activated HRasG12V increased the fraction of cells that have acquired epithelial phenotype and express EpCAM, particularly in KTB42 “,this stament is only true for one sample out of 3, in the other 2 samples this behavior is in fact over the overexpression of mutant HRasG12V . many of the results are related only with one sample, not Evan the majority of the samples,

As I already mentioned, one of my main objections is that the results, as expected due to the very limited number of samples evaluated, are really heterogeneous. The authors sometimes described the anecdotal result of one PZP cell sample and generalized the observed phenotype to all the phenomena. For example: “Transformation of PZP cells with activated HRasG12V increased the fraction of cells that have acquired epithelial phenotype and express EpCAM, particularly in KTB42”, this statement is only true for one sample out of 3, in the other 2 samples this behavior is in fact, opposite. Many of the results are related to only one sample, but presented as the general phenotype.

Response: We agree with the reviewer these were from one cell line in our previous submission. We now report similar observation with KTB40 as well and results are shown in Figure 8. In addition, these results were obtained without transformation with RAS oncogene. Thus, parental PZP cells without RAS transformation can undergo transdifferentiation when in contact with epithelial cells. Furthermore, transdifferentiated cells acquire ALDEFLUOR-positivity, which is shown in Figure S10. Moreover, we have reproduced transdifferentiation properties of RAS transformed KTB42 cells *in vivo*, as bulk PROCR+/EpCAM- or single cell derived clones from PROCR+/EpCAM- cells generated tumors with islands of EpCAM+ tumor cells (Figure S15).

23. Describe results regarding SV40-T/t antigen and their combination

Response: Thank you for pointing out this error. We have included additional details (page 21) and included reference to our earlier publication with more details (Molecular Cancer Research 19: 1802-1817).

24. Include y axis title in Fig Sup 1

Response: Thank you. We have made necessary modifications.

25. Indicate the total numbers of evaluated samples in each plot

Response: Figure legends have been modified to include number of samples.

26. Be more moderate in generalizing this phenomenon to women with AA ancestry, since their strategy does not allow their conclusions to be extended to the population level

Response: Thank you for the suggestion. We have modified the manuscript accordingly.

Reviewer 4:

1. Use of the term Latina or Latino ancestry is confusing. Specifically, Table 1 includes race in the title but then goes on to include Hispanic/Latino.

Response: We used the 940 subjects from HGDP project [PMID: 18292342, 15803201] as a reference to infer the continental origins. The reference dataset included subjects from seven continental regions, Africa, the Middle East, Europe, Central/South Asia, East Asia, the

Americas and Oceania. Among those, 64 Native Americans were from 3 geographic origins and 5 subpopulations: Mexico (Prima, Maya), Colombia (Colombians), and Brazil (Surui, Karitiana).

The races reported in the manuscript are genetic ancestries. We agree with the reviewer and replaced Latina with Indigenous American in the manuscript.

2. Both introduction and discussion are lacking proper contextualization of the study within the large amount of research that has been conducted investigating higher risk of breast cancer mortality among African American/Black women. The foundation for this paper should better acknowledge the work done by others demonstrating that disparities affecting AA/Black women have not been fully explained and that prior research has shown biological differences.

Response: We sincerely apologize for this error. We have included additional references while adhering to the journals requirement of ~70 references. References 3, 5, 6, 12-15 are new in the revised manuscript

3. The panel of 41 SNPs does not ‘discern’ the continental origins of participants, it provides an estimation of ancestry proportions.

Response: We agree with the reviewer that the admixture proportions are inferred.

Further specification of parameters set for the process of estimating ancestry proportions is needed.

Response: STRUCTURE (version 2.3.4) was run with standard parameter settings, and the HGDP was used as predefined reference populations with 7 inferred genetic clusters. The admixture model with prior information on sample origins for reference panel was applied with 50,000 burn-in steps and 100,000 MCMC iterations.

4. For ancestry estimation with the small number of 41 aims authors used 1000G data. However, ideally, they should have used a large non-admixed reference set **as the authors of the original 41-aim panel paper described.**

Response: We used 940 subjects from HGDP project as a reference to infer the continental origins and admixture proportions. The subjects were from 52 global populations and seven

geographical regions. It is the **same reference panel used by the 41 AIMs publication** (PMID: 23815888).

5. Authors should provide mean and SD measure for ancestry estimates.

“Potential issues in the ancestry estimations in the current article are reflected by the high Central South Asia estimates for the Hispanic/Latina samples. The average Central South Asia estimate for Hispanic/Latinos is usually very low (ranging from 1 to 4%), and lower than the Indigenous American component of ancestry (in their paper referred to 'American'). Authors should have provided mean and SD measures for ancestry estimates instead of just figures.”

Response: The statistics for inferred proportion of ancestry for each group of individuals are summarized below,

Reference Population	Black (N=48)	Hispanic (N=45)	White (N=171)
Africa			
Minimum	0.3030	0.0020	0.0020
1st Qu	0.6405	0.0060	0.0030
Median	0.7325	0.0090	0.0050
Mean	0.7392	0.0515	0.0089
3rd Qu	0.8575	0.0200	0.0080
Maximum	0.9860	0.8790	0.1840
SD	0.1579	0.1457	0.0166
America			
Minimum	0.0010	0.0010	0.0010
1st Qu	0.0038	0.0180	0.0030
Median	0.0070	0.0770	0.0050
Mean	0.0136	0.1959	0.0089
3rd Qu	0.0120	0.3890	0.0085
Maximum	0.1440	0.6040	0.0930
SD	0.0237	0.2016	0.0125
Central/South Asia			
Minimum	0.0030	0.0060	0.0060
1st Qu	0.0168	0.0380	0.0140
Median	0.0370	0.1170	0.0250
Mean	0.0633	0.2001	0.0573
3rd Qu	0.0623	0.2670	0.0500
Maximum	0.4240	0.8240	0.8520
SD	0.0854	0.2154	0.1009
East Asia			
Minimum	0.0020	0.0020	0.0020
1st Qu	0.0060	0.0090	0.0040

Median	0.0100	0.0140	0.0060
Mean	0.0150	0.0405	0.0091
3rd Qu	0.0170	0.0340	0.0100
Maximum	0.0840	0.2760	0.0690
SD	0.0157	0.0631	0.0100
Europe			
Minimum	0.0030	0.0300	0.0840
1st Qu	0.0245	0.1240	0.6780
Median	0.0595	0.2420	0.8550
Mean	0.0763	0.3465	0.7686
3rd Qu	0.1105	0.5320	0.9255
Maximum	0.2540	0.9580	0.9740
SD	0.0635	0.2765	0.2161
Middle Est			
Minimum	0.0030	0.0150	0.0080
1st Qu	0.0340	0.0600	0.0265
Median	0.0595	0.1320	0.0600
Mean	0.0828	0.1525	0.1409
3rd Qu	0.1190	0.2080	0.1935
Maximum	0.3540	0.4910	0.7700
SD	0.0747	0.1183	0.1732
Oceania			
Minimum	0.0020	0.0020	0.0020
1st Qu	0.0030	0.0040	0.0030
Median	0.0050	0.0070	0.0040
Mean	0.0098	0.0130	0.0064
3rd Qu	0.0095	0.0130	0.0070
Maximum	0.0860	0.1450	0.0460
SD	0.0141	0.0220	0.0071

Barplot of ancestry proportion.

- Variation in ancestry proportions—consider reanalyzing data looking at correlations between marker and African ancestry proportion.

Response: The correlation between marker genes and African ancestry proportion are listed below. Two genes are positively correlated with African ancestry proportions and all three genes

are negatively correlation with European ancestry proportions. These results are discussed in pages 11 and 12,

Pearson correlation coefficient and p-value

	PDGFR_H_Score		PROCR_H_Score		ZEB1_H_Score	
	coefficient	p-value	coefficient	p-value	coefficient	p-value
Africa	0.347	2.75E-06	0.484	1.62E-10	0.065	0.40
America	-0.120	0.11	-0.022	0.79	0.217	4.34E-03
Central/South Asia	0.082	0.28	0.083	0.30	0.254	8.16E-04
East Asia	0.060	0.43	0.046	0.57	0.112	0.15
Europe	-0.265	4.00E-04	-0.404	1.74E-07	-0.208	6.45E-03
Middle Est	-0.084	0.27	-0.113	0.16	-0.020	0.80
Oceania	0.040	0.60	0.249	1.75E-03	0.035	0.65

REVIEWER COMMENTS

Reviewer #2 (Remarks to the Author):

In this revision, the authors have largely addressed my previous concerns.

One clarification is needed: lines 481-2 (page 22), “KTB42-HRasG12V cell-derived tumor was ER α -/GATA3-/FOXA1- (Fig. 10D)”. However, the IHC picture in Fig. 10D clearly showed that this tumor had positive nuclear staining for FOXA1, so it’s FOXA1+ rather than FOXA1-. In contrast, the KTB42 cell line transformed with BOTH mutant HRasG12V and SV40-T/t antigen had negative staining for FOXA1 (Fig. 10E). Please clarify.

Reviewer #3 (Remarks to the Author):

The authors answered most of my comments, but I still have a few questions:

1. Overall, the authors conclusively demonstrated that PZP cells have properties of fibroadipogenic/mesenchymal stromal cells that express PROCR and PDGFR α and transdifferentiate into adipogenic and osteogenic lineages. They also described the enrichment of this cellular compartment in normal tissues from AA women. Luminal cells also enhanced the acquisition of basal cell characteristics after cocultivation with PDZ cells. But I'm not really convinced by the argument that PDZ cells are one of the cells of origin of metaplastic breast cancers. Let me explain my point, by definition metaplastic breast carcinomas are triple-negative breast cancers, and in fact, the molecular alterations that distinguish metaplastic from triple-negative cancer are poorly understood. As I already commented in my first report, I do not see a clear relationship, particularly with metaplastic cancer and why it cannot be extended to other triple negative cancers, mainly claudin-low tumors, as described by the author for Zeb1. The authors state that the relationship of the PDZ cell population to oncogenesis is that higher basal levels of PZP cells in the breast tissue of AA women could promote tumor growth, in my opinion this could partially explain the high incidence of triple-negative tumors in AA patients . Unfortunately, from clinical table S1, it is not clear how many non-metaplastic triple negative tumors were evaluated. First, I would ask to change the individualized report of ER, PR and Her2 to the combination of HR/HER2 markers, it would be clearer in this way to define how many tumors of each subtype were evaluated.

The authors reported that phospho-STAT3, a surrogate marker of IL-6 activity and PDZ presence, was over-expressed in normal and cancerous breast tissues from AA women, but none of the tumors tested

were metaplastic. Therefore, the only evidence to conclude that stromal PZP enrichment can explain the higher incidence of metaplastic breast cancer in Afro-descendant women are experiments with mouse models and the KTB42 cell line, none of the data presented demonstrated this in human tumors, and its exclusive relationship with the metaplastic tumor type. I am not sure if public data such as PXD014414 or GSE57549 could be explored to better describe the association with metaplastic human tumors, or TCGA RPPA data including STAT3_pY705 in 937 samples from non-Hispanic white and AA women. This can be explored in conjunction with RNA-seq data to explore IL-6 and TAGLN in NAT and tumor compartments in breast cancer, as well as tumor subtypes. I mean, the present results confidently describe the existence of PDZ compartment, their role in normal mammary cells, and the changes associated with ancestry, but I cannot robustly arrive, with the actual data, at the big conclusion presented: the PDZ enrichment could explain the higher incidence of metaplastic breast cancer in women with a higher contribution of African ancestry. I think a more detailed description of human samples should be allowed to suggest this more strongly.

2. I think the term Indigenous American Ancestry is not particularly correct, we generally report the Mesoamerican and South American components as Native American ancestry (Mexican and South American indigenous population). In the actual form, in addition to Mexican, Central American and South American indigenous groups, native populations from Canada and the United States could be included, both none of these groups were evaluated.

Reviewer #4 (Remarks to the Author):

I appreciate the effort from the authors to respond to all of the reviewers' comments. Despite some improvement, I believe that the authors need to better define the constructs of race/ethnicity and genetic ancestry and be consistent in their use throughout their manuscript and their experiments.

The approach used by the authors regarding nomenclature to refer to either self-reported race/ethnicity or genetic ancestry is still confusing throughout the manuscript. A large proportion of the women in the Komen 'Hispanic' category group do not have Indigenous American ancestry, so using that term to refer to that group is erroneous and misleading. Authors should define the construct they use and apply it appropriately depending on what they are showing and discussing. If they conduct analyses stratified by self-reported race/ethnicity category based on the Komen intake survey, then they should be clear about this and use the right category that people are choosing from when they respond (e.g., 'Hispanic'). When they conduct analyses by genetic ancestry, then they can identify individuals with some Indigenous American ancestry in the 'Hispanic' category and refer to those participant as 'individuals with Indigenous American ancestry'. Similarly, if they are focusing on individuals with African or European ancestry influences, then they can refer to those participants as 'individuals with African

ancestry' or 'individuals with European ancestry'. Those groups would include different people compared to groups defined by the Komen survey response regarding race/ethnicity self-report.

The genetic ancestry proportions of the Komen 'Hispanic' individuals are different to those that have been reported for Hispanic/Latino individuals in the US and individuals in Latin America. This category usually includes individuals of European, African, and Indigenous American ancestries, with smaller proportion of Asian influence. Based on the data presented, the 'Hispanic' individuals in Komen have on average 20% Asian and 20% Indigenous American ancestry influence, which is not consistent with all previous reports for self-identified 'Hispanics'. I am unsure of why this is the case, but it is apparent that the individuals in the 'Hispanics' category of the Komen tissue repository, or at least the 45 included in this particular analysis, should not be referred to as 'Indigenous American'. And any conclusions based on analyses of such a heterogeneous group of individuals in terms of their ancestry should be avoided.

I have two other minor comments:

2. The introduction now includes more information about previous studies exploring potential factors leading to higher breast cancer mortality in African American/Black women compared to non-Hispanic White women. However, I would recommend that the authors review the flow and the quality of the writing in this section, specially the first two sentences. Expressions such as 'much higher' or 'even ...' could be avoided.

3. In my previous comments I wrote "The samples used in the study seem to have variation in ancestry proportions. Hence, the authors could have conducted studies using samples with similar proportion of African and European ancestry, independently of race/ethnic category of the participant and contrast results using the categories". I did not see a response to this comment in the Rebuttal letter.

Response to Review:

Reviewer #2 (Remarks to the Author):

In this revision, the authors have largely addressed my previous concerns.

Response: Thank you for your positive comment.

One clarification is needed: lines 481-2 (page 22), “KTB42-HRasG12V cell-derived tumor was ER α -/GATA3-/FOXA1- (Fig. 10D)”. However, the IHC picture in Fig. 10D clearly showed that this tumor had positive nuclear staining for FOXA1, so it’s FOXA1+ rather than FOXA1-. In contrast, the KTB42 cell line transformed with BOTH mutant HRasG12V and SV40-T/t antigen had negative staining for FOXA1 (Fig. 10E). Please clarify.

Response: We apologize for this error. We went back to our original data and reconfirmed FOXA1 expression in KTB42HRasG12V cell-derived but not HRASG12V+SV40-T/t antigen-derived tumors. We have corrected this error (page 24 of the revised manuscript).

Reviewer #3 (Remarks to the Author):

The authors answered most of my comments, but I still have a few questions:

1. Overall, the authors conclusively demonstrated that PZP cells have properties of fibroadipogenic/mesenchymal stromal cells that express PROCR and PDGFR α and transdifferentiate into adipogenic and osteogenic lineages. They also described the enrichment of this cellular compartment in normal tissues from AA women. Luminal cells also enhanced the acquisition of basal cell characteristics after cocultivation with PDZ cells. But I’m not really convinced by the argument that PDZ cells are one of the cells of origin of metaplastic breast cancers. Let me explain my point, by definition metaplastic breast carcinomas are triple-negative breast cancers, and in fact, the molecular alterations that distinguish metaplastic from triple-negative cancer are poorly understood. As I already commented in my first report, I do not see a clear relationship, particularly with metaplastic cancer and why it cannot be extended to other triple negative cancers, mainly claudin-low tumors, as described by the author for Zeb1. The authors state that the relationship of the PDZ cell population to oncogenesis is that higher basal levels of PZP cells in the breast tissue of AA women could promote tumor growth, in my opinion this could partially explain the high incidence of triple-negative tumors in AA patients . Unfortunately, from clinical table S1, it is not clear how many non-metaplastic triple negative tumors were evaluated. First, I would ask to change the individualized report of ER, PR and Her2 to the combination of HR/HER2 markers, it would be clearer in this way to define how many tumors of each subtype were evaluated.

Response: Thank you for your insightful comments. We hope explanation provided in the revised manuscript addresses above comments. In our opinion, PZP cells contributes to breast tumorigenesis in two different ways. First, as a source of IL-6 to promote luminal cells acquiring

basal cell gene expression and thus predisposing luminal breast epithelial to an aggressive transformed phenotype. With basal gene expression, tumors resulting from such cells are inherently triple negative. We suspect this is the major contribution of PZP cells in aggressive TNBCs in African American women. The second is that PZP cells themselves as one of cells-of-origin of metaplastic cancers. This project was not initiated with identifying cells of origin of metaplastic cancers. We transformed a series of epithelial and PZP cell lines generated from healthy breast tissues with defined oncogenes (SV40-T/t antigens and HRAS^{G12V} or PIK3CA^{H1047R}) just to determine what types of tumors these transformed cells generate. While epithelial cell lines generated adenocarcinomas or squamous carcinomas (Molecular Cancer Research 19:1802-1817), PZP cells generated only metaplastic carcinomas. It is possible that metaplastic carcinomas generated by PZP cells are claudin-low subtype as described by Weigelt et al. (Modern Pathology 28:340-351). In this respect, analysis of previous RNA-seq data comparing gene expression in immortalized PZP cells compared immortalized epithelial cells (Cancer Research 78:5107-5123) showed lower expression of CDH1, CLDN7 (no expression of CLDN3 and CLDN4) in PZP cells compared to epithelial cells. Lack or reduced expression of CDH1, CLDN3, CLDN4 and CLDN7 is the unique feature of Claudin-low breast cancers. Additional support for PZP cells being cell-of-origin of metaplastic carcinoma is based on our observation of elevated expression of EMT-associated genes such as ZEB1, ZEB2, SNAI1, and TWIST1 in PZP cells compared to epithelial cells. Metaplastic breast cancers in general express higher levels of EMT-associated genes compared to other TNBC subtypes (Nature Communications 11:1723). GSEA analysis of genes differentially expressed in PZP cells compared to epithelial cells also revealed enrichment of EMT, IFN γ , Apical junction signatures but reduced OXPPOS signatures in PZP cells, which is similar to pathways found specifically in metaplastic carcinomas compared to other TNBCs described by Djomehri et al (Nature Communications 11:1723 (please see below for comparison). Since cell-of-origin gene signatures instead of mutation driven signatures predominate in 33 cancer types that have been examined (Cell 173:291-304), these results further support the possibility of PZP cells being cells-of-origin of metaplastic carcinomas. These explanations are included in results section (pages 25-26) and GSEA profiles of PZP cells that overlap with that of metaplastic carcinomas are shown in Figure S14E.

As suggested, Table S1 has been modified to include additional details of tumors used.

The authors reported that phospho-STAT3, a surrogate marker of IL-6 activity and PDZ presence, was over-expressed in normal and cancerous breast tissues from AA women, but none of the tumors tested were metaplastic. Therefore, the only evidence to conclude that stromal PZP enrichment can explain the higher incidence of metaplastic breast cancer in Afro-descendant women are experiments with mouse models and the KTB42 cell line, none of the data presented demonstrated this in human tumors, and its exclusive relationship with the metaplastic tumor type. I am not sure if public data such as PXD014414 or GSE57549 could be explored to better describe the association with metaplastic human tumors, or TCGA RPPA data including STAT3_pY705 in 937 samples from non-Hispanic white and AA women. This can be explored in conjunction with RNA-seq data to explore IL-6 and TAGLN in NAT and tumor compartments in breast cancer, as well as tumor subtypes. I mean, the present results confidently describe the existence of PDZ compartment, their role in normal mammary cells, and the changes associated with ancestry, but I cannot robustly arrive, with the actual data, at the big conclusion presented: the PDZ enrichment could explain the higher incidence of metaplastic breast cancer in women with a higher contribution of African ancestry. I think a more detailed description of human samples should be allowed to suggest this more strongly.

Response: Thank you for these suggestions and accordingly we have analyzed suggested data. As explained above, PZP cells are enriched for metaplastic carcinoma enriched gene signatures. We first examined RPPA data of 403 tumors described in the TCGA data set (Nature 490:61). Although STAT3-Y705 was measured in that study, phospho-STAT3 levels did not show breast cancer subtype-specific differences. We then used UALCAN database (Neoplasia 25:18-27) to explore CPTAC data (Nature 534:55) for phosphorylation status of STAT3 in breast cancers based on race. This dataset had only 9 metaplastic tumor samples and therefore, metaplastic breast cancer specific changes in STAT3 status could not be verified. Results of other analyses are shown below. Multiple sites in STAT3 are phosphorylated and phosphorylation of several of these sites are higher in tumors from self-reported African American/Black and non-Hispanic white women compared to Asian women. Since these data were generated from self-reported race, we are unsure of their relevance to data presented in the manuscript and, therefore, we did not elaborate further in the manuscript. Also, sample size is small to find a meaningful difference between self-reported African American/black versus non-Hispanic white.

To complement above results, we examined the TCGA dataset for STAT3 gene signature (BMC Medicine 13:177). Consistent with phospho-STAT3 data, expression levels of STAT3 signature genes were higher in tumors from African American/Black and non-Hispanic white women compared to Asian women. Since number of PZP cells increase in tumor microenvironment compared to normal breast in case of non-Hispanic white (Figures 4-6), whereas phospho-STAT3 is intrinsically higher in the normal breast of African American/Black women (Figure 9), it is possible that there is tumor-specific elevation of STAT3 signatures in non-Hispanic white as well, although at lower level than in tumors of African American women (Figure 9).

We also explored TCGA RNA-seq data for IL6 and TAGLN. mRNA counts for IL6 were too low to draw conclusions. However, mRNA counts for TAGLN were high enough for the analysis and the expression was significantly higher in mesenchymal stem-like subtype of TNBCs compared to other TNBCs such as TNBC-LAR and TNBC-IM. Mesenchymal stem-like subtypes of TNBCs show similarity to Claudin-low and EMT-enriched breast cancers.

2. I think the term Indigenous American Ancestry is not particularly correct, we generally report the Mesoamerican and South American components as Native American ancestry (Mexican and South American indigenous population). In the actual form, in addition to Mexican, Central American and South American indigenous groups, native populations from Canada and the United States could be included, both none of these groups were evaluated.

Response: Thank you for the suggestion. We used guidelines outlined in JAMA (326:621) as suggested by the editor. However, these guidelines did not give specific recommendation on terms to be used to describe Hispanic population. We have changed nomenclature to Native American ancestry as suggested here.

Reviewer #4 (Remarks to the Author):

I appreciate the effort from the authors to respond to all of the reviewers' comments. Despite some improvement, I believe that the authors need to better define the constructs of race/ethnicity and genetic ancestry and be consistent in their use throughout their manuscript and their experiments.

The approach used by the authors regarding nomenclature to refer to either self-reported race/ethnicity or genetic ancestry is still confusing throughout the manuscript. A large proportion of the women in the Komen 'Hispanic' category group do not have Indigenous American ancestry, so using that term to refer to that group is erroneous and misleading. Authors should define the construct they use and apply it appropriately depending on what they are showing and discussing. If they conduct analyses stratified by self-reported race/ethnicity category based on the Komen intake survey, then they should be clear about this and use the right category that people are choosing from when they respond (e.g., 'Hispanic'). When they conduct analyses by genetic ancestry, then they can identify individuals with some Indigenous American ancestry in the 'Hispanic' category and refer to those participant as 'individuals with Indigenous American ancestry'. Similarly, if they are focusing on individuals with African or European ancestry influences, then they can refer to those participants as 'individuals with African ancestry' or 'individuals with European ancestry'. Those groups would include different people compared to groups defined by the Komen survey response regarding race/ethnicity self-report.

Response: Thank you for insightful comments. We have clarified this throughout the manuscript. As recommended, African American and non-Hispanic white groups indicate self-reported via questionnaire data at the time of tissue collection, while African ancestry or European ancestry is used when describing classification based on ancestry-informative markers. We have removed the women who self-reported "Hispanic" ethnicity based on inconsistencies with the ancestry-informed groups identified in an earlier review (see below for more detail).

The genetic ancestry proportions of the Komen 'Hispanic' individuals are different to those that have been reported for Hispanic/Latino individuals in the US and individuals in Latin America. This category usually includes individuals of European, African, and Indigenous American ancestries, with smaller proportion of Asian influence. Based on the data presented, the 'Hispanic' individuals in Komen have on average 20% Asian and 20% Indigenous American ancestry influence, which is not consistent with all previous reports for self-identified 'Hispanics'. I am unsure of why this is the case, but it is apparent that the individuals in the 'Hispanics' category of the Komen tissue repository, or at least the 45 included in this particular analysis, should not be referred to as 'Indigenous American'. And any conclusions based on analyses of such a heterogeneous group of individuals in terms of their ancestry should be avoided.

Response: We have taken your advice, reviewer #2 comments and Editors advice on ancestry reporting guidelines published in JAMA and proposed to use the term Native American ancestry for these 45 samples. We also did additional analysis by excluding 7 samples, which are enriched for Asian ancestry. Results are similar to what we observed with the entire group, as shown below. Since results are same, we have described these results in the manuscript without removing original data (pages 13-14).

Compare H-score and Positivity between genetic ancestry mapped tissues from women of African, European and Native American ancestry for PROCR

	Genetic Ancestry				
variable label	COLUMN_OVERALL	African N=31	European N=129	Native American N=26	P-value
Positivity	0.17 (0.04, 0.55)	0.30 (0.06, 0.55)	0.15 (0.04, 0.41)	0.17 (0.04, 0.32)	<.0001
H-Score	27.92 (5.45, 127.65)	56.41 (10.46, 127.65)	24.37 (6.68, 83.50)	24.61 (5.45, 58.45)	<.0001

Note: Values expressed as median (min, max)

Note: P-value comparisons across race categories are based on Kruskal-Wallis

Compare H-score and Positivity between genetic ancestry mapped tissues from women of African, European and Native American ancestry for ZEB1

	Genetic Ancestry				
variable label	COLUMN_OVERALL	African N=33	European N=144	Native American N=22	P-value
Positivity	0.01 (0.00, 0.24)	0.01 (0.00, 0.10)	0.01 (0.00, 0.12)	0.01 (0.00, 0.24)	0.1455
H-Score	1.55 (0.15, 30.72)	2.21 (0.24, 16.47)	1.46 (0.15, 19.76)	1.83 (0.40, 30.72)	0.0380

Note: Values expressed as median (min, max)

Note: P-value comparisons across race categories are based on Kruskal-Wallis

Compare H-score and Positivity between genetic ancestry mapped tissues from women of African, European and Native American ancestry for PDGFR α

	Genetic Ancestry				
variable label	COLUMN_OVERALL	African N=35	European N=154	Native American N=13	P-value
Positivity	0.10 (0.01, 0.75)	0.28 (0.03, 0.75)	0.09 (0.01, 0.73)	0.08 (0.02, 0.35)	<.0001
H-Score	16.00 (1.43, 110.23)	36.96 (6.25, 110.23)	14.27 (1.43, 88.61)	9.43 (2.69, 52.62)	<.0001

Note: Values expressed as median (min, max)

Note: P-value comparisons across race categories are based on Kruskal-Wallis

I have two other minor comments:

2. The introduction now includes more information about previous studies exploring potential factors leading to higher breast cancer mortality in African American/Black women compared to non-Hispanic White women. However, I would recommend that the authors review the flow and the quality of the writing in this section, specially the first two sentences. Expressions such as ‘much higher’ or ‘even ...’ could be avoided.

Response: Thank you, the introduction, particularly the first paragraph, has been significantly revised. We utilized the terminology for race and ethnicity used in the cited manuscripts, and have also noted that...”These studies have all relied on self-reported race and ethnicity, which are not biologically-defined, but are instead social constructs that have changed over time.” We then go on to describe studies that have used ancestry rather than race.

3. In my previous comments I wrote “The samples used in the study seem to have variation in ancestry proportions. Hence, the authors could have conducted studies using samples with similar proportion of African and European ancestry, independently of race/ethnic category of the participant and contrast results using the categories”. I did not see a response to this comment in the Rebuttal letter.

Response: We apologize for not providing details in our previous rebuttal letter. The original selection of samples into the study population was based on self-reported race and ethnicity, not ancestry-informative markers. We have revised our analysis, eliminating the samples from women who self-reported Hispanic ethnicity (but, as pointed out earlier, did not have ancestry markers that aligned with this report) and re-analyzed the remaining samples as noted in above tables. Number of samples varied based on markers due to drop-off from TMA. We agree with the reviewer that creating TMA by selecting samples from those with highest African or European ancestry would be ideal. However, it is difficult to put that in practice as we are working with limited tissue materials with variable cellularity and with very high drop-off with each cut of the TMA. However, we did another analysis by comparing markers in tissues with >75% African or European ancestry as shown below. Overall results did not change much (except for ZEB1, which was not significant, possibly due to low sample size) and results of these analyses are included in the revised manuscript (pages 12-14).

Compare H-score by Ancestry for PROCR

	group			
variable label	COLUMN_OVERALL	>75% African N=17	>75% European N=70	P-value

	group			
variable label	COLUMN_OVERALL	>75% African N=17	>75% European N=70	P-value
PROCR H Score	27.86 (7.49, 113.00)	56.41 (10.46, 113.00)	21.97 (7.49, 64.70)	<.0001

Note: Values expressed as median (min, max)

Note: P-value comparisons across race categories are based on Wilcoxon (Normal Approximation)

Compare H-score by Ancestry for PDGFR

	group			
variable label	COLUMN_OVERALL	>75% African N=16	>75% European N=87	P-value
PDGFR H Score	14.50 (1.43, 88.61)	38.62 (2.69, 87.74)	13.41 (1.43, 88.61)	0.0027

Note: Values expressed as median (min, max)

Note: P-value comparisons across race categories are based on Wilcoxon (Normal Approximation)

Compare H-score by Ancestry for ZEB1

	group			
variable label	COLUMN_OVERALL	>75% African N=13	>75% European N=83	P-value
ZEB1 H-Score	1.37 (0.15, 30.72)	1.66 (0.27, 13.83)	1.37 (0.15, 30.72)	0.9658

Note: Values expressed as median (min, max)

Note: P-value comparisons across race categories are based on Wilcoxon (Normal Approximation)

REVIEWERS' COMMENTS

Reviewer #3 (Remarks to the Author):

In general, the authors significantly improved the discussion and description of their findings. Most of my concerns were well addressed, and therefore I believe the article is suitable for publication.

Reviewer #4 (Remarks to the Author):

I appreciate the effort that the authors put into using consistent racial/ethnic construct definitions and distinguishing between analyses based on genetic ancestry estimates and analyses based on racial/ethnic self-identified categories.

I noticed however that the authors made the choice of not including the ancestry range for the participants in the Hispanic self-reported category in the results section (they do include explicit ancestry ranges for the NHW and African American/Black categories) but they then go ahead and write "Genetic ancestry marker distribution patterns in self-reported Hispanic women showed expected genetic admixture." This is not actually the case, as when one inspects Supplementary figure 4 it is clear that the ancestry proportions for the Hispanics are not consistent with those described in previous studies. If they know of other studies in Hispanics reporting similar ancestry distributions to those in their analyses, they should include a citation to provide evidence for the assertion.

Given the clear heterogeneity (beyond usually described patterns) in the group that self-identified as Hispanic, I would not write that the distribution is expected and instead I would honestly describe the heterogeneity for this category. I would suggest to the author to also work on the ancestry estimation process again. A total of 41 markers is relatively small, and the allele frequencies for the multiple pseudo ancestral populations included might not be differentiated enough.

Response to review:

Reviewer #3 (Remarks to the Author):

In general, the authors significantly improved the discussion and description of their findings. Most of my concerns were well addressed, and therefore I believe the article is suitable for publication.

Response: Thank you for your favorable recommendation.

Reviewer #4 (Remarks to the Author):

I appreciate the effort that the authors put into using consistent racial/ethnic construct definitions and distinguishing between analyses based on genetic ancestry estimates and analyses based on racial/ethnic self-identified categories.

I noticed however that the authors made the choice of not including the ancestry range for the participants in the Hispanic self-reported category in the results section (they do include explicit ancestry ranges for the NHW and African American/Black categories) but they then go ahead and write "Genetic ancestry marker distribution patterns in self-reported Hispanic women showed expected genetic admixture." This is not actually the case, as when one inspects Supplementary figure 4 it is clear that the ancestry proportions for the Hispanics are not consistent with those described in previous studies. If they know of other studies in Hispanics reporting similar ancestry distributions to those in their analyses, they should include a citation to provide evidence for the assertion.

Given the clear heterogeneity (beyond usually described patterns) in the group that self-identified as Hispanic, I would not write that the distribution is expected and instead I would honestly describe the heterogeneity for this category. I would suggest to the author to also work on the ancestry estimation process again. A total of 41 markers is relatively small, and the allele frequencies for the multiple pseudo ancestral populations included might not be differentiated enough.

Response: Considering the concerns and issues with genetic ancestry analysis, we removed the Hispanic/Latinx samples from our analyses. Details are as follow– Abstract, Page 3 (Line 51,52); Result section, Page 11-14 (Line 209, 212-213, 219-224, 236, 242-244, 252-253, 257-259, 272, 277-279); Methods, Page 36-38, (Line 753-757, 796-797, 820); Figure legends, Page 55-56 (Line 1302, 1304-1307, 1312-1313,1320, 1322-1323, 1334, 1336-1338, 1342). All the data was re-analyzed considering AA vs EA population, P-values are updated in the Figures 4, 5, and 6. The statistical test used are updated accordingly. Table S2 has been updated with corresponding values. In the Supplementary files, figure legends for figure S6, S7, and S8 have been updated (all the data related to the NA population have been removed). P-values are updated in Figures S6, S7, and S8. Figure S4 has been updated. For convenience, we have attached the marked version of the manuscript.